# Mechanism of activation at the selectivity filter of the KcsA K+ channel

**Florian T Heer[1,2], David J Posson[3,4], Wojciech Wojtas-Niziurski[1,2], Crina M Nimigean[3,4,5]\*, Simon Bernèche[1,2]\***

[1]SIB Swiss Institute of Bioinformatics, University of Basel, Basel, Switzerland; [2]Biozentrum, University of Basel, Basel, Switzerland; [3]Department of Anesthesiology, Weill Cornell Medical College, New York, United States; [4]Department of Physiology and Biophysics, Weill Cornell Medical College, New York, United States; [5]Department of Biochemistry, Weill Cornell Medical College, New York, United States

**Abstract** Potassium channels are opened by ligands and/or membrane potential. In voltage-gated K+ channels and the prokaryotic KcsA channel, conduction is believed to result from opening of an intracellular constriction that prevents ion entry into the pore. On the other hand, numerous ligand-gated K+ channels lack such gate, suggesting that they may be activated by a change within the selectivity filter, a narrow region at the extracellular side of the pore. Using molecular dynamics simulations and electrophysiology measurements, we show that ligand-induced conformational changes in the KcsA channel removes steric restraints at the selectivity filter, thus resulting in structural fluctuations, reduced K+ affinity, and increased ion permeation. Such activation of the selectivity filter may be a universal gating mechanism within K+ channels. The occlusion of the pore at the level of the intracellular gate appears to be secondary.

DOI: https://doi.org/10.7554/eLife.25844.001

**\*For correspondence:**
crn2002@med.cornell.edu (CMN);
simon.berneche@unibas.ch (SB)

**Competing interests:** The authors declare that no competing interests exist.

## Introduction

In voltage-gated potassium channels, the opening of an intracellular gate formed by a 'bundle-crossing' of the inner pore-lining helices is hypothesized to allow K ions to flow through the pore, down their electrochemical gradient (*Yellen, 1998*). A narrow section of the pore located towards the extracellular side of the membrane, called the selectivity filter because it allows the pore to select for K ions, has been proposed to play double duty as an inactivation gate: following a stimulus, the pore opens at the intracellular bundle-crossing and, in the continued presence of the stimulus, the selectivity filter changes its conformation to prevent further flux of K ions. Thus, in voltage-gated K+ channels, the intracellular bundle-crossing of the pore-lining helices is believed to be the activation gate, while the selectivity filter is believed to be the inactivation gate. This gating mechanism is believed to be shared by a subset of K+ channels such as KcsA and inward rectifier channels (*Phillips and Nichols, 2003*; *Cuello et al., 2010a*; *Cuello et al., 2010b*; *Uysal et al., 2011*). However, functional experiments examining state-dependent accessibility of pore blockers (*Contreras and Holmgren, 2006*; *Wilkens and Aldrich, 2006*; *Rapedius et al., 2012*; *Posson et al., 2013b*; *Posson et al., 2015*) or thiol-modifying agents (*Flynn and Zagotta, 2001*; *Zhou et al., 2011*), have shown that in other channels the intracellular pore-lining helices no longer form a bundle crossing that obstructs K+ flux in the closed state, thus strongly implicating the selectivity filter as an activation gate (*Flynn and Zagotta, 2001*; *Proks et al., 2003*; *Bruening-Wright et al., 2007*; *Klein et al., 2007*; *Contreras et al., 2008*). This suggests that different K+ channels with high sequence and structural homology do not share a universal gating mechanism. Here, we challenge this statement based on our findings with KcsA, a structurally characterized model K+ channel, which was believed to activate by simple opening of the

**eLife digest** Potassium channels are proteins found in almost all living organisms and are vital for many different biological processes. These proteins contain a pore that allows potassium ions to flow through cell membranes, but only when the channel is open. Most channels have a narrowing at the inward side of the pore, which was proposed to form a gate that controls the flow of ions. This gate only opens when the channel activates. Nearer the outward side of the pore is another narrow region called the selectivity filter. This region interacts selectively with potassium ions as they pass through the channel.

Not all channels form a tight constriction at the inward end of their pore, yet they still only allow potassium ions to flow through when activated. This suggested that these channels might instead use the selectivity filter as a gate. It would also mean that whilst different potassium channels have similar structures, they do not share a common gating mechanism that controls the flow of ions.

Heer et al. studied the bacterial channel called KcsA, which was thought to control the flow of potassium ions via a traditional inward gate. Computer simulations based on this protein's structure, and experiments with purified KcsA in artificial membranes, showed that the selectivity filter was also involved when KcsA was activated. When the activated channel changed shape, the selectivity filter – which had been constrained by regions forming the pore – could now move and allow potassium ions to flow through the pore. Heer et al. confirmed using a mutant KcsA that the motion of the inward side of the pore upon activation affected the movement of the selectivity filter gate as predicted by the simulation.

These findings show that the KcsA channel opens and closes at the selectivity filter. The changes at the inward side of the pore, previously believed to be the gate, firstly enable the selectivity filter to serve as a gate while also forming a secondary gate. Heer et al. propose that most if not all potassium channels may also use this mechanism. These findings illustrate that molecular simulations can be powerful for predicting how changes in the structure of a protein will affect its behavior. This and future studies of other potassium channels will help scientists to better understand the subtle differences between the diverse range of channels found across different organisms.
DOI: https://doi.org/10.7554/eLife.25844.002

intracellular bundle-crossing gate. We find that this outward movement of the transmembrane helices in KcsA leads to a change in the selectivity filter from closed to conductive during channel activation. We thus propose that the selectivity filter is the main gate in KcsA.

Ion permeation through the selectivity filter of K$^+$ channels has been investigated and described based on KcsA X-ray crystallography data (*Morais-Cabral et al., 2001*; *Zhou et al., 2001*) and molecular mechanics simulations (*Aqvist and Luzhkov, 2000*; *Bernèche and Roux, 2001*). The generally accepted model for permeation involves alternating states of the selectivity filter loaded with two or three ions separated by water molecules, low ion desolvation free energy at the filter entryway, and the occurrence of a knock-on transitional state in which two K ions come in close contact (*Bernèche and Roux, 2001*). Structures of KcsA are available with the inner helices in different configurations, making KcsA a good model to investigate ion permeation gating (*Cuello et al., 2010a*; *Uysal et al., 2011*). In the 'closed' KcsA structure, the inner helices are closed at the bundle crossing gate (*Doyle et al., 1998*; *Zhou et al., 2001*), while in the 'activated' KcsA crystal structures, the inner helices are wide open at the bundle crossing (*Cuello et al., 2010b*). Here, we analyze and compare the mobility of K$^+$ ions within the selectivity filter in both the 'closed' and 'activated' states of KcsA.

## Results

### Relaxation of the selectivity filter is required for ion permeation

We performed free energy calculations in order to describe ion permeation through the selectivity filter of the 'closed' KcsA channel, which is thought to be stabilized in a conductive state (pdb entry 1K4C) (*Zhou et al., 2001*). The plot in *Figure 1A* shows a projection of the potential of mean force (PMF) as a color-coded energy map that varies as a function of the position of the three K ions along the channel axis. The reduced reaction coordinate $Z_{12}$ corresponds to the center-of-mass of the two

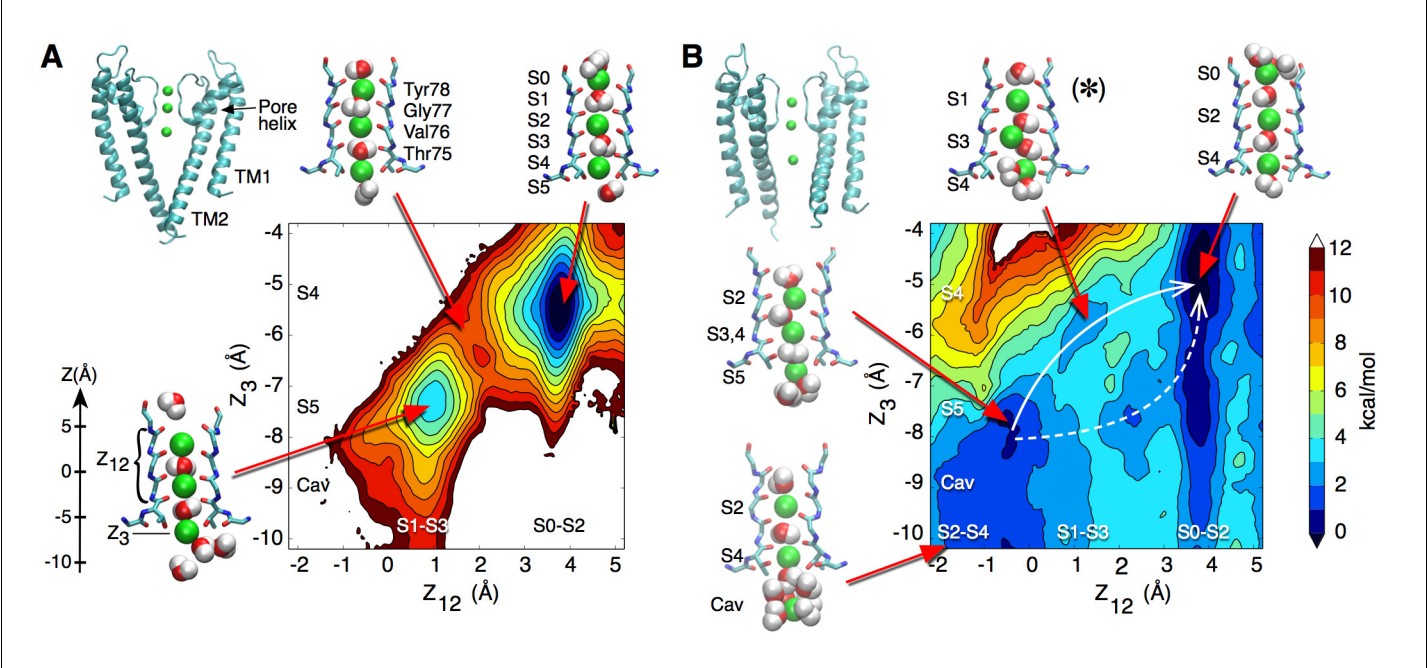

**Figure 1.** Potential of mean force calculations describing ion permeation through the selectivity filter in the closed and open intracellular gate conformations of the channel. (**A**) In the closed conformation (based on structure 1K4C), ions are tightly bound to the selectivity filter in states S1-S3-S5 or S0-S2-S4 with little ion movement due to high free energy barriers > 6 kcal/mol (**B**) In the open conformation (based on structure 3F5W), ion diffusion is possible due to low free energy barriers of 2–3 kcal/mol. A knock-on transition state (S1–S3–S4) is observed (identified by (*)). The reaction coordinate $Z_{12}$ corresponds to the center-of-mass of the two outermost ions, and $Z_3$ to the lower ion. Ion positions are defined relative to the center-of-mass of the backbone atoms of the selectivity filter core residues (Thr75-Val76-Gly77-Tyr78). Molecular structures illustrate two of the four channel subunits and for the key states, K ions and water molecules (green and red/white spheres, respectively) are shown with the selectivity filter backbone of residues Thr74 to Gly79.

DOI: https://doi.org/10.7554/eLife.25844.003

The following figure supplements are available for figure 1:

**Figure supplement 1.** Potential of mean force describing ion permeation in the selectivity filter of the closed KcsA channel starting from the S1-S3-Cav occupancy state.
DOI: https://doi.org/10.7554/eLife.25844.004

**Figure supplement 2.** Modeling of the open-activated selectivity filter based on open-inactivated structures.
DOI: https://doi.org/10.7554/eLife.25844.005

**Figure supplement 3.** Potential of mean force describing ion permeation in the open channel.
DOI: https://doi.org/10.7554/eLife.25844.006

**Figure supplement 4.** Standard deviation for the PMFs presented in *Figure 1*.
DOI: https://doi.org/10.7554/eLife.25844.007

ions on the extra-cellular side, and $Z_3$ to the innermost ion, relative to the center of mass of the selectivity filter. Thus, one can read directly from the plot the relative free energy of the filter configuration corresponding to any position of the three ions. As examples, the molecular structures of the key ion occupancy states corresponding to well and saddle points along the permeation pathway are illustrated. The PMF calculation, which was initiated with K ions in sites S0-S2-S4 separated by water molecules in S1 and S3, shows deep free energy wells of at least 6 kcal/mol, indicating high K$^+$ binding affinity (*Figure 1A*). In disagreement with the assumption of a conducting filter, these calculations suggest that ion permeation is rather impeded. Our result did not depend on the initial ion configuration, as a PMF calculation started with K ions in sites S1-S3 and Cav (cavity) with water molecules in S2 and S4, reveals similar free energy wells about 7 kcal/mol deep (*Figure 1—figure supplement 1*). Unlike the first PMF, this one shows the formation of a knock-on state achieved via the loss of the water molecule between the 2nd and 3rd ions. Similar occupancy states were observed by other groups who proposed mechanisms where ions in the filter are most often not separated by water molecules during permeation, the higher ion density resulting in greater ionic repulsion and

current (*Furini and Domene, 2009*; *Köpfer et al., 2014*). However, comparison of the PMFs in *Figure 1A* and *Figure 1—figure supplement 1* shows that the loss of a water molecule did not significantly reduce the free energy barriers in our calculations.

We sought a different explanation for the slow $K^+$ diffusion. We wondered whether the barriers to permeation we calculated were due to a non-permissive conformation of the KcsA selectivity filter, adopted when the channel's intracellular activation gate is 'closed'. To explore this possibility, we next examined the free energy of ion movement through the selectivity filter of a KcsA channel with an 'open' intracellular gate. All KcsA 'open' states documented in the literature allow $K^+$ access through the intracellular entryway but display a selectivity filter in a collapsed state, in agreement with functional data showing that KcsA inactivates within seconds after activation by protons (*Chakrapani et al., 2011*). Permeation is prohibited through such a collapsed filter conformation as rotation of the amide planes within the collapsed selectivity filter disrupt K ion binding and lead to new, stabilizing interactions of the amide planes with water molecules that intercalate between the subunits (*Figure 1—figure supplement 2A*). Removal of these water molecules has been shown to facilitate recovery from inactivation and restore the 'conductive' form of the selectivity filter (*Ostmeyer et al., 2013*).

Two KcsA structures with an open bundle-crossing and collapsed selectivity filter (pdb entries 3F7V and 3F5W, with intracellular entryway diameters of 23 Å and 32 Å, respectively) (*Cuello et al., 2010b*) were chosen for the preparation of simulations of an open KcsA channel. In line with the study of *Ostmeyer et al., 2013*, we prepared a conductive selectivity filter conformation using a simulation in which the tightly bound water molecules behind the selectivity filter were removed and K ions maintained at the canonical binding sites S1-S3-Cav or S0-S2-S4 using harmonic restraints (see Materials and methods). No direct restraints were applied to the selectivity filter itself in order to minimize the possibility of introducing artifactual structural changes in the channel. This procedure led to a selectivity filter nearly identical in structure with that of the closed state KcsA (*Figure 1—figure supplement 2B*). The PMF calculation describing ion movement within the selectivity filter of the open state KcsA, based on the 3F5W structure and initiated with ions in sites S0-S2-S4, reveals small free energy barriers of 2 to 3 kcal/mol, conducive to ion permeation (*Figure 1B*). A PMF describing a complete permeation event (the entrance of a K ion on one side and the release of a K ion from the other side) is shown in *Figure 1—figure supplement 3*. Compared to the above simulations using the closed KcsA structure (*Figure 1A*), these results indicate that a dramatic increase in K ion conductance occurs within the selectivity filter after the channel has switched from a conformation with the intracellular gate closed to one where it is open.

Ion permeation in the open state appears to be favored by increased fluctuations and slight enlargement of the selectivity filter. The histogram in *Figure 2A*, extracted from 20-ns long simulations of the open and closed states, shows that following the opening of the intracellular gate the selectivity filter adopts conformations that are up to 1 Å wider in diameter. This slight widening of the selectivity filter allows for a knock-on transition state in which two K ions accompanied by a water molecule (as opposed to one $K^+$ and one water) occupy the S3 and S4 binding sites (state identified by (*) in *Figure 1B*). The knock-on state is not strictly required for permeation and an alternative pathway, referred to as vacancy-diffusion, is also accessible (*Figure 1B*, dashed arrow). The vacancy-diffusion pathway involves that ions occupying the selectivity filter move first, leaving a vacant site that is filled by an incoming ion (*Schumaker and MacKinnon, 1990*; *Bernèche and Roux, 2001*).

Interestingly, in a previous simulation work, permeation via both the knock-on and vacancy-diffusion pathways was observed despite the fact that those calculations were performed using a structure of KcsA in which the intracellular gate is closed (*Bernèche and Roux, 2001*). In those simulations, the fluctuations of the selectivity filter required for ion permeation were originating from the intrinsic flexibility of the protein main chain, which was overestimated by CHARMM22, a previous generation of the CHARMM force field (see section 'Force field and ion permeation' in Materials and methods). For comparison, the distance distribution in the selectivity filter of the closed state channel is also plotted for a simulation using the CHARMM22 force field (*Figure 2A*, dashed line). The plot shows fluctuations similar to those observed in the open channel using the current CHARMM36 force field. This new generation of force field notably contains a correction term for backbone dihedral angles developed on the basis of X-ray crystallographic data and quantum mechanical calculations of the protein backbone conformational energy (*Mackerell et al., 2004*;

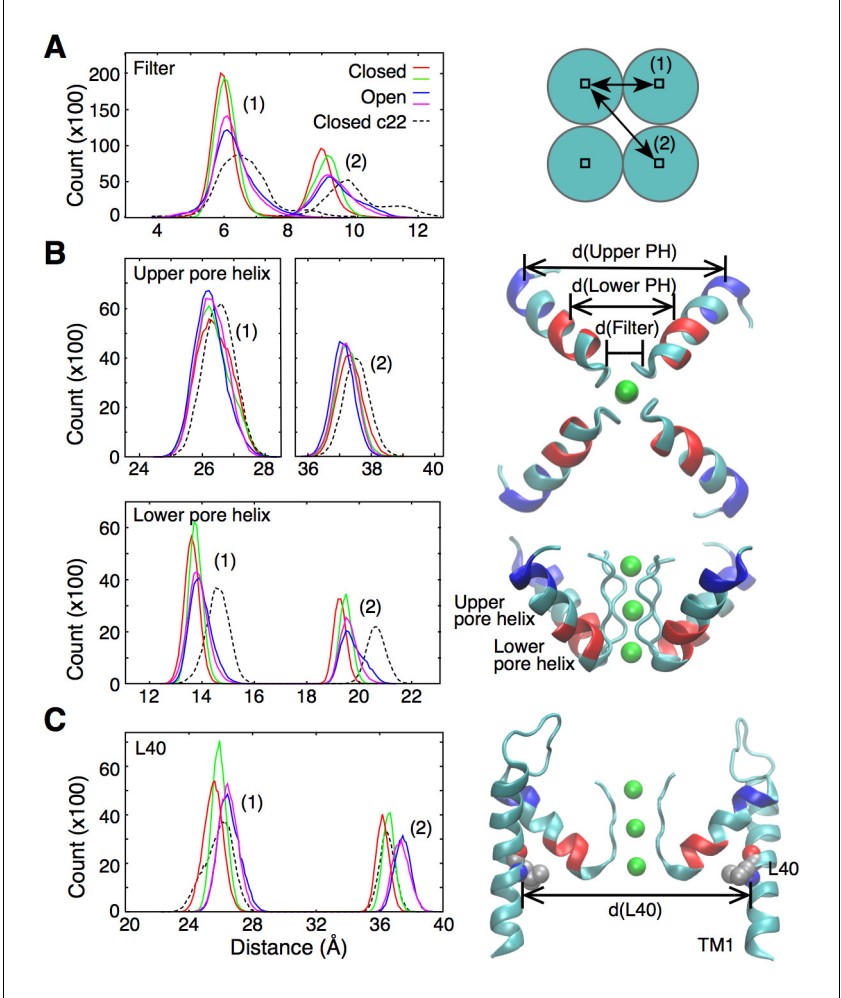

**Figure 2.** Fluctuations around the selectivity filter. (**A**) The histogram shows the inter-subunit, adjacent (1) and opposite (2), distances between the $C_\alpha$ atoms of residues 75 to 78. In the open state of the channel (two independent simulations shown in blue and pink), the selectivity filter can reach conformations that are wider by up to 1 Å in comparison to the closed state (red and green lines). Simulations of the closed channel using the CHARMM22 force field resulted in broader distance distributions (dashed lines) than those obtained with CHARMM36 force field in both the closed and open states. (**B**) The inter-subunit distances between the center-of-mass of the $C_\alpha$ atoms of the upper pore helix (residues 62 to 65) show little changes between the closed and open states of the channel. By contrast, the distances increase by up to 1 Å at the level of the lower pore helix (residues 70 to 73). The closed channel with the CHARMM22 force field also displays longer distances at the level of the lower pore helix. Bottom and side views of the selectivity filter and the pore helix (PH) are shown with the upper and lower pore helix segments colored in blue and red, respectively. K ions are shown in green. (**C**) The histogram shows that the inter-subunit distances between the $C_\alpha$ atoms of the L40 residues, increase by about 1 Å upon opening of the intracellular gate. The CHARMM22 force field has no impact on the L40 distances in the closed channel. For both the open and closed states, two independent 20-ns long simulations were analyzed. The CHARMM22 simulation data were taken from *Bernèche and Roux (2001)*.
DOI: https://doi.org/10.7554/eLife.25844.008

*Best et al., 2012*). The improved force field allows us to better understand how the fluctuations of the selectivity filter are actually controlled by allosteric interactions involving the transmembrane and pore helices.

Departing from the permeation mechanism described here where three ions, accompanied by waters, permeate the selectivity filter, *Köpfer et al., 2014* suggested a hard knock-on mechanism in which ion permeation arises from the entrance of a fourth ion, excluding at the same time water

molecules from the selectivity filter. The discrepancy between the two proposed permeation mechanisms is in part due to differences in the force field parameters defining the $K^+$-carbonyl interaction, which result in much stronger ion binding affinity to the selectivity filter in the study by Köpfer et al. (see section 'Force field and ion permeation' in Materials and methods). However, it is important to consider that, as shown here, the ion binding affinity is essentially determined by the functional state of the selectivity filter. Simulations performed by *Jensen et al. (2013)* suggest that the activated state of the channel, which has the lowest ion binding affinity, is not susceptible to the effect of ion parameters. Their simulations of the Kv1.2/2.1 chimeric channel have shown that the ion force field has little influence on the permeation mechanism since parameters that yield different magnitudes of $K^+$-carbonyl interaction still sustained similar mechanisms involving alternating ions and water molecules (*Jensen et al., 2013*). It remains unclear why permeation in the open KcsA channel appears to be susceptible to the ion force field while permeation in the Kv1.2/2.1 chimera is not (*Jensen et al., 2013*; *Köpfer et al., 2014*).

## The outer helix is involved in activation gating

As shown in *Figure 2B*, the widening of the selectivity filter is associated with movement of the pore helix. Interestingly, only the lower part of the pore helix moves outward upon activation, while the upper part of the pore helix remains mostly unchanged. The lower pore helix segment is in direct contact with the outer transmembrane helix (TM1) at the level of residue L40 (*Figure 3*). The side chain of this leucine is within van der Waals distance from the side chains of Ser69 and Val70 at the bottom of the pore helix. The simulations reveal that the distances between the L40 alpha carbon ($C_\alpha$) atoms of opposing subunits increases by about 1 Å as the TM1s move outward upon opening of the intracellular gate (*Figure 2C*). It is this outward movement of TM1 at the level of L40 that gives the pore helix and the selectivity filter the required room to expand at the base and become conductive. This mechanism predicts that reducing the volume of the residue at position 40 by mutation to a smaller amino acid would similarly provide more space for the pore helix and selectivity filter to expand, and would make it easier for the filter to open and become conductive. Such a mutation would thus favor permeation by reducing the coupling between channel activation and selectivity filter opening. To test this hypothesis, we performed electrophysiological analysis of the L40A KcsA channel mutant.

KcsA is a pH-dependent channel that activates upon binding of protons to at least two pH-sensor residues, H25 and E118 (*Figure 4A*) (*Thompson et al., 2008*; *Posson et al., 2013a*). We performed single channel recording in lipid bilayers to evaluate the effect of the L40A mutation on the pH-dependent gating of KcsA. For these experiments, we employed a routinely-used KcsA variant where inactivation was removed via the previously described E71A mutation (*Cordero-Morales et al., 2006*; *Thompson et al., 2008*; *Posson et al., 2013a*). *Figure 4B* shows that the L40A mutant is fully activated by protons, similar to the E71A control channel (open probability increases from 0 to 1 upon pH change), and with a similar pH at half activation (pH$_{0.5}$=5.2, see *Table 1*). However, the mutant displayed a shallower proton dose response compared with the control channel, indicating a markedly decreased sensitivity to protons (*Figure 4B*), characterized by a lower Hill coefficient (Hill coefficient is ~2 for L40A compared to ~4.5 for the control channel, see *Equation 1*, *Table 1*). Importantly, the L40A mutant channel opened at a much lower proton concentration than the control channel (pH 6, *Figure 4B–C*) and displayed intermediate activity over a range of proton concentrations, unlike the very steep $H^+$-dependent activation displayed by the control channel (*Figure 4C*), suggestive of a channel whose opening is less strongly coupled to proton binding.

In order to interpret the effects of the L40A mutant on KcsA gating, we analyzed the pH-dependence with a model we previously used to characterize the effects of mutations of channel residues directly involved in proton-sensing (*Posson et al., 2013a*). The gating behavior displayed by L40A was identical to that of H25R, a mutant designed to mimic constitutive protonation at histidine 25, the KcsA proton sensor where proton binding is most strongly coupled to channel opening (*Figure 4B–C*) (*Posson et al., 2013a*). First, the mutation of H25 to arginine removed the proton binding sites at H25 that strongly and cooperatively contributed to the opening of the control channel, yielding a shallower dose-response for the mutant channel (*Figure 4B*). Second, the arginine substitution mimicked histidine protonation, which dramatically increased the intrinsic gating equilibrium ($L_o$) towards the open state (*Equation 2*, *Table 1*). Surprisingly, the L40A mutation displays a

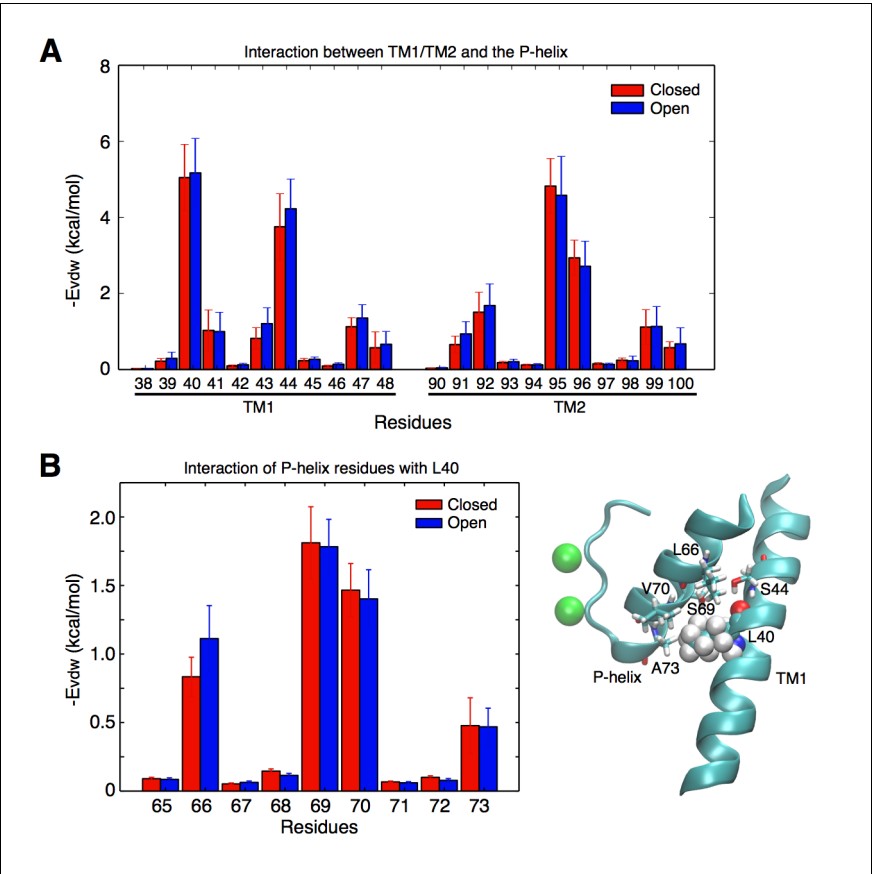

**Figure 3.** Contacts between the TM helices and the pore helix that can potentially transmit the activation signal. (**A**) vdW interactions of the individual residues of the TM1 and TM2 helices with the pore helix. Residue Leu40 from TM1 forms the strongest interactions. Interactions involving Ser44 are also important, but this residue forms an intra-helix H-bond with the backbone of Leu40 and thus was not mutated. The interaction of Leu40 with the pore helix is along an axis directed toward the permeation pore, and can thus potentially impact on the pore size. Residues from TM2 also interact with the pore helix, but these interactions are not directed toward the pore, and thus were not further investigated. (**B**) Leu40 interacts with the bottom of the pore helix, mainly at the level of residues Ser69, and Val70.

DOI: https://doi.org/10.7554/eLife.25844.009

proton dose-response that is as shallow as H25R (*Figure 4B*), suggesting that the potency of the H25 sensor is diminished in this mutant by as much as removing the sensor altogether. In addition, the L40A channel begins to open at lower proton doses in much the same way as H25R (*Figure 4C*). We thus propose that the L40A mutation similarly favors channel opening both by increasing the intrinsic gating ($L_o$) compared to the E71A control and by altering the state-dependence of H25 protonation (see Materials and methods and *Table 1*). Both changes are necessary to fit the data and constitute the most parsimonious model to fit the gating behavior of the L40A mutant. We propose that these effects result from partial uncoupling of the bottom of the pore helix from the TM1 helix and from the protonation state of H25. A residue at a position equivalent to KcsA L40 was previously found to control permeation and open probability in another K channel, a $Ca^{2+}$-activated K channel (KCa3.1) that gates at the selectivity filter (*Garneau et al., 2014*), suggesting a common mechanism.

To verify the impact of the L40A mutation on $K^+$ permeation, we performed simulations of this mutant using the closed conformation (pdb entry 1K4C), as well as the 3F7V structure in which the main gate is open less widely than in 3F5W (*Cuello et al., 2010b*). For consistency with the electrophysiology experiments described above, these simulations used the E71A inactivation-removed mutant as control. The free energy calculations presented in *Figure 5* combine data from independent automated umbrella sampling simulations initiated in different ion occupancy states, as detailed

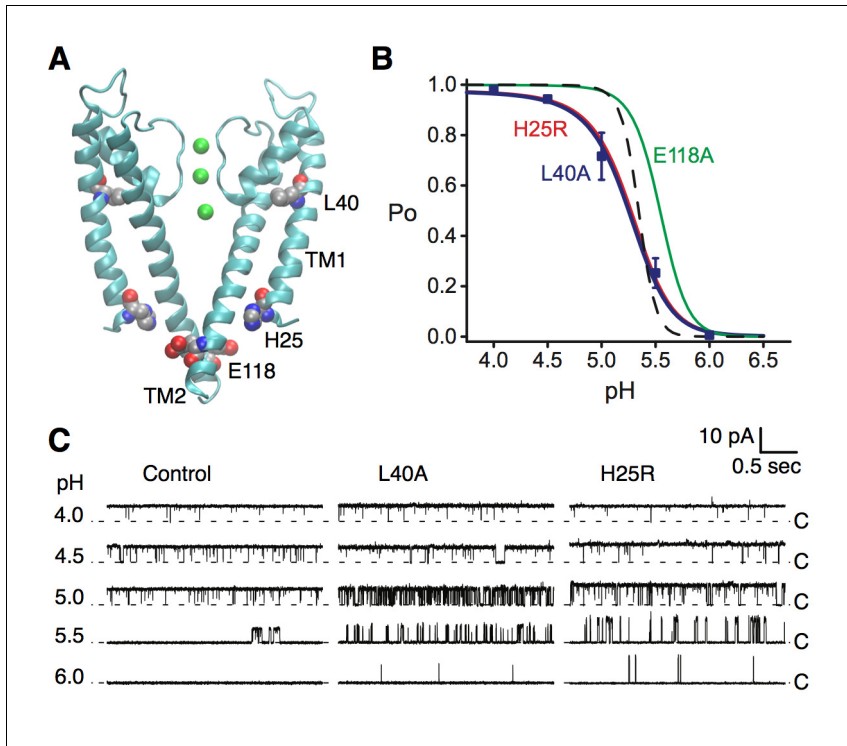

**Figure 4.** Coupling of the selectivity filter to the intracellular gate via residue L40. (**A**) The pH sensor residues, H25 and E118, are found on the intracellular side of the channel. Residues H25 and L40 both lie on the transmembrane helix TM1. (**B**) KcsA L40A open probability (Po) vs. pH from single channel recording (blue symbols) was fit to a model for pH-dependent gating (*Equation 1*, blue line). Data are the mean from 4 or 5 bilayers ± sem. Previously reported dose response curves (*Thompson et al., 2008*) for the E71A control channel (dashed line), pH sensor mutants H25R (red) and E118A (green) are provided for comparison (*Table 1*). (**C**) Representative single channel traces at pH values 4 to 6 and 100 mV for the E71A (Control, left), E71A/L40A (center), and E71A/H25R (right) channels, illustrating that the L40A mutant, similar to H25R, opens at lower [H$^+$] (pH 6 and 5.5) compared to E71A control. Traces were filtered offline (100 Hz) for display. The identity of the gates that open and close during the single-channel recordings is unknown.

DOI: https://doi.org/10.7554/eLife.25844.010

**Table 1.** Summary of Hill and pH-gating model fits for the KcsA mutants indicated.

| Mutant* | Hill fit | | pH sensor 1 (H25) | | pH sensor 2 (E118) | | Intrinsic gating |
|---|---|---|---|---|---|---|---|
| | pH$_{1/2}$ (±) | n$_H$ (±) | pK$_{a1}$$^{closed}$ (±) | pK$_{a1}$$^{open}$ (±) | pK$_{a2}$$^{closed}$ (±) | pK$_{a2}$$^{open}$ (±) | Lo (±) |
| E71A[†] control channel | 5.3 (0.01) | 4.4 (0.1) | 4.8 (N/A)[‡] | 7.6 | 5.0 | 6.2 | 2.5E-12 (0.3E-12) |
| L40A | 5.2 (0.02) | 1.9 (0.2) | 7.6[§] | 7.6 | 5.0 | 6.2 | 6.2E-4 (0.7E-4) |
| H25R[†] | 5.3 (0.02) | 1.9 (0.2) | - | - | 5.0 (0.1) | 6.2 (0.1) | 7E-4 (4E-4) |
| E118A[†] | 5.5 (0.03) | 4.5 (2.8) | 4.8 | 7.6 | - | - | 1.1E-8 (0.1E-8) |

*All mutants on the background of E71A.

[†]Data and fits from Refs. (*Thompson et al., 2008*) and (*Posson et al., 2013a*).

[‡]Not applicable; no error given when the parameters were constrained (see Materials and methods).

[§]The data were fit by abolishing the H25 pK$_a$ state-dependence, similar to the H25R mutant.

DOI: https://doi.org/10.7554/eLife.25844.011

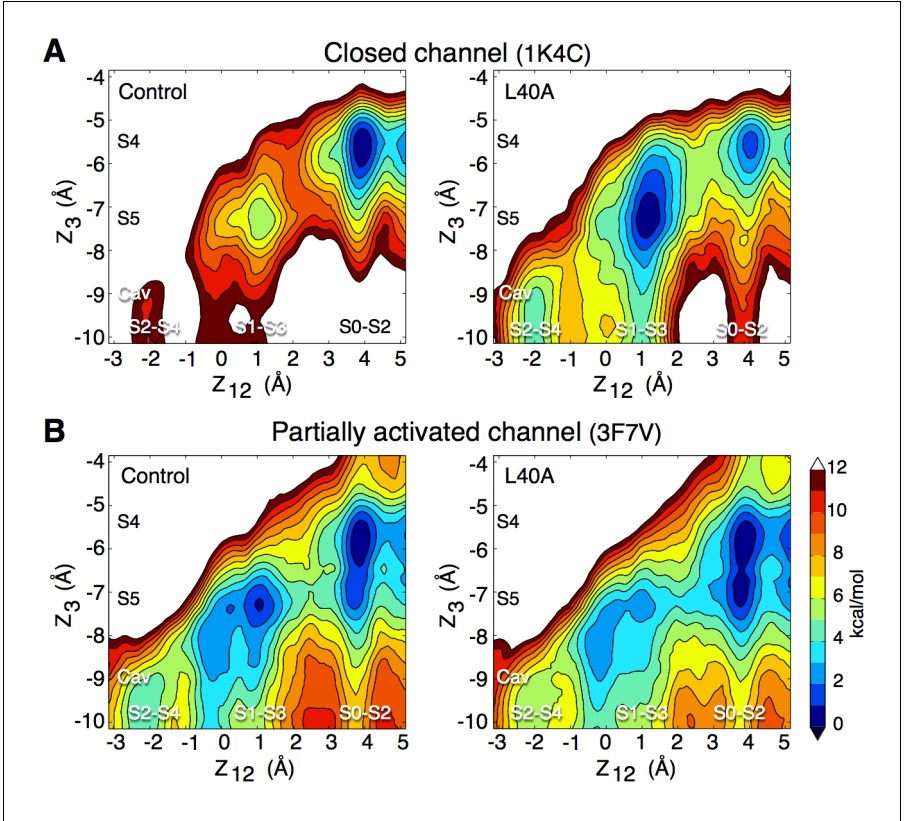

**Figure 5.** Impact of the L40A mutation on ion permeation in the closed and partially activated channel. (**A**) The PMF calculations show that when the intracellular gate is closed (pdb entry 1K4C), ion diffusion is impeded in the control channel (E71A) with a free energy difference between the 3-ion (S0–S2–S4) and the 2-ion (S1-S3-Cav and S2-S4-Cav) states of about 11 kcal/mol, and free energy barriers of up to 9 kcal/mol between these states. The L40A mutation stabilizes the 2-ion states by about 6 kcal/mol and reduces the free energy barriers to 4–5 kcal/mol. (**B**) In the partially activated structure (pdb entry 3F7V), both constructs show reduced free energy barriers of 4 kcal/mol for the control and 2–3 kcal/mol for the L40A mutant. Each of the 4 PMFs shown here combines together data from two independent automated umbrella-sampling simulations initiated in states S1-S3-Cav and S0-S2-S4, respectively. Since the statistical error on the PMFs combined in this fashion might not be readily definable, we only present the statistical errors associated with the underlying individual PMFs (*Figure 5—figure supplement 1, 4* and *5*). Despite this limitation, the combined PMFs shown here provide a convenient summary of the underlying calculations presented separately in *Figure 5—figure supplement 1*.

DOI: https://doi.org/10.7554/eLife.25844.012

The following figure supplements are available for figure 5:

**Figure supplement 1.** Forward and backward PMFs of the transition between the S1-S3-Cav and S0-S2-S4 ion occupancy states in the control and L40A mutant channels for (**A**) the closed (1K4C) and (**B**) the partially activated (3F7V) conformations.

DOI: https://doi.org/10.7554/eLife.25844.013

**Figure supplement 2.** Potential of mean force calculation describing ion permeation in the E71A mutant in the fully open conformation (pdb entry 3F5W).

DOI: https://doi.org/10.7554/eLife.25844.014

**Figure supplement 3.** Ion permeation in the partially activated channel with the L40A mutation.

DOI: https://doi.org/10.7554/eLife.25844.015

**Figure supplement 4.** Convergence of the potential of mean force calculations.

DOI: https://doi.org/10.7554/eLife.25844.016

**Figure supplement 5.** Standard deviation for the PMFs presented in *Figure 5—figure supplement 1*.

DOI: https://doi.org/10.7554/eLife.25844.017

in *Figure 5—figure supplement 1*. The PMFs of both *Figure 5* and *Figure 5—figure supplement 1* lead to the following observations. In the closed conformation (1K4C) of the control channel, the state with 3 ions bound to the filter (S0-S2-S4) is overly stabilized in comparison to states with only 2 ions bound to the filter (S1-S3-Cav and S2-S4-Cav), with a free energy difference of about 11 kcal/mol. Free energy barriers of 6 to 10 kcal/mol are observed between the 2-ion and 3-ion states. The E71A/L40A mutant brings about fluctuations that reduce the ion binding affinity and the relative stability of the 3-ion state. The 2-ion states are stabilized by about 6 kcal/mol, and are thus more accessible. The free energy barriers along the permeation pathway are also reduced to 4–5 kcal/mol. In the partially open conformation (3F7V), the control channel shows reduced free energy barriers of about 4 kcal/mol. The L40A mutant further reduces the barriers by 1 to 2 kcal/mol, to 2–3 kcal/mol. For comparison, a PMF describing ion permeation in the E71A mutant of the fully open channel (3F5W) shows barriers of 3 kcal/mol (*Figure 5—figure supplement 2*).

A simulation of the E71A/L40A mutant in the partially activated state (3F7V) was performed with an applied transmembrane voltage of 400 mV and an ion concentration of 800 mM KCl. The time-series analysis in *Figure 5—figure supplement 3* shows permeation events in which at most three ions are bound to the selectivity filter with accompanying water molecules, in agreement with the above PMFs and with previously proposed permeation models (*Bernèche and Roux, 2001*; *Morais-Cabral et al., 2001*). These results support the idea that the L40A mutation facilitates the activation of the channel by reducing the extent of the required conformational change and associated energy to reach the maximal open probability.

## Discussion

Our work on KcsA revealed a high ion binding affinity state of the selectivity filter that corresponds to a resting (closed) state in which the filter is stabilized, non-conducting but primed for ion conduction, while the intracellular gate is closed. Upon activation, in addition to the large motion of the inner TM2 helix that opens the intracellular gate, the reorientation of the outer TM1 helix releases steric restraints at the selectivity filter allowing ion conduction (*Figure 6A*). This is in agreement with the observation made by EPR experiments that the conducting state of the KcsA selectivity filter displays larger fluctuations than non-conducting states (*Raghuraman et al., 2014*). Such changes in the structural dynamics of the selectivity filter through activation could also explain the sub-conductance levels with altered selectivity detected by electrophysiology measurements in Shaker (*Chapman et al., 1997*; *Zheng and Sigworth, 1997*; *Chapman and VanDongen, 2005*). Our work shows how fluctuations at the selectivity filter have direct impact on its conductance and gating properties, and could potentially be affected by other signaling factors, e.g. temperature and membrane tension (*Rodríguez et al., 1998*; *Clarke et al., 2010*; *Dong et al., 2015*).

The central role played by the outer TM1 helix in KcsA activation potentially explains why residue H25, found at the intra-cellular end of TM1, was identified as the strongest pH sensor (*Posson et al., 2013a*). The corresponding helix in voltage-dependent channels is S5, which is directly connected to the voltage-sensor through a linker (S4-S5 linker). This linker is believed to interact directly with the S6 helix to open the channel at the intracellular 'bundle crossing'. While S5 was generally thought to play little role in activation gating, it is plausible that the voltage-sensor exerts force on S5 and by doing so regulates the conductance of the selectivity filter through interactions with the pore helix, without necessarily engaging the S6 helix via the S4-S5 linker (*Lees-Miller et al., 2009*; *Garg et al., 2013*; *Garneau et al., 2014*). In support of this hypothesis, previous results by other groups showed that voltage activation can occur without an S4-S5 linker (*Lörinczi et al., 2015*) and that the S4-S5 linker in a subset of voltage-dependent channels does not directly engage the S6 helix (e.g. EAG channels (*Whicher and MacKinnon, 2016*)). This does not exclude that the inner TM helices may form, in KcsA and other channels (such as voltage-gated channels), a second steric barrier at the intracellular end of the pore. This model also fits the behavior of $Ca^{2+}$-activated $K^+$ channels, found to ultimately open and close at the selectivity filter, although the gating is accompanied by a large movement of the pore-lining inner helices not unlike that found in voltage-gated channels and KcsA, albeit less obstructive at the intracellular entryway (*Posson et al., 2015*).

In conclusion, our work contributes a unified gating mechanism for all $K^+$ channels. We propose that the selectivity filter can be found in at least three different states that determine the functional states of the channel: a restrictive closed state, a dynamic and slightly expanded open state, and a

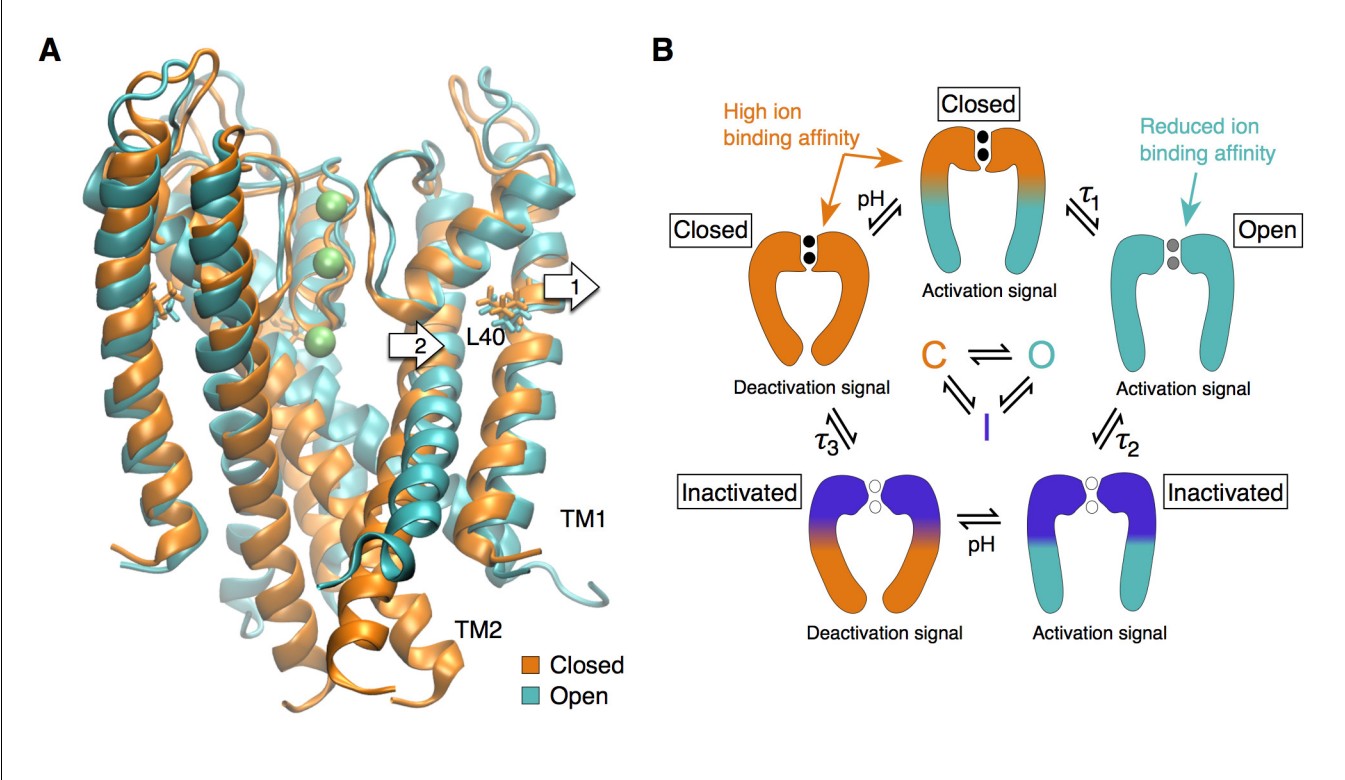

**Figure 6.** The selectivity filter as the center of the activation mechanism. (**A**) The superposition of the open and closed channel illustrates how the movement of the TM1 helices (arrow 1) allows for the displacement of the pore helix and slight expansion of the selectivity filter (arrow 2). (**B**) The selectivity filter is found in three states (closed (resting), open, inactivated) that determine the functional state of the channel. The TM helices transmit to the selectivity filter the activation signal coming from a ligand- or voltage-dependent domain. The crossing of the TM helices on the intra-cellular side of the pore can form a steric barrier, but is not strictly essential to the function of K channels.
DOI: https://doi.org/10.7554/eLife.25844.018

pinched inactivated state (*Figure 6B*). Transitions between these functional states are controlled by the channel transmembrane domains that provide distant allosteric interactions between the selectivity filter and disparate regulatory sites such as voltage sensor and ligand binding domains (*Panyi and Deutsch, 2006*; *Clarke et al., 2010*; *Cuello et al., 2010a*; *Wylie et al., 2014*; *Posson et al., 2015*). Thus, we propose that the universal gate in K[+] channels is the selectivity filter, whereas the pore-lining helices may undergo large motions that can lead to a secondary closure point at the intracellular bundle-crossing for a subset of K[+] channels.

## Materials and methods

### Molecular simulation systems

The molecular membrane systems were built based on the X-ray structure of the KcsA channel in its high-[K[+]] closed conformation (pdb entry 1K4C) (*Zhou et al., 2001*) and the open inactivated structures (pdb entries 3F7V and 3F5W) (*Cuello et al., 2010a*). Helical turns were added to the transmembrane helices of the two open structures so that these helices have the same length in all constructs, each subunit containing residues 22 to 124. All residues were assigned their standard protonation state at pH 7, except Glu71, which was protonated. The E71A and L40A mutations were applied to the X-ray structures and a new system was built for each independent case. The systems were assembled using the CHARMM-GUI web-service (*Jo et al., 2008*) following a protocol developed by Woolf and Roux (*Woolf and Roux, 1994*; *Wu et al., 2014*). The protein channel, with its symmetry axis aligned along the Z-axis, was embedded in a lipid bilayer of about 130 dipalmitoyl-phosphocholine (DPPC) molecules in the case of WT, or dioleoylphosphocholine (DOPC) for E71A

constructs. The number of ions in the bulk was adjusted to reproduce an ionic concentration of about 150 mM KCl and to obtain neutral systems, which typically yields a total of 19 cations and 31 anions. The resulting molecular systems each contained about 48,000 atoms.

All calculations were performed using the CHARMM software version c36 (*Brooks et al., 2009*). The all-atom potential energy function CHARMM36 (*Best et al., 2012*) was used for protein and phospholipids, and water molecules were modeled using the TIP3P potential (*Jorgensen et al., 1983*). The Lennard-Jones (NBFIX) parameters for the $K^+$-carbonyl oxygen pair interactions were refined so that the solvation free energy of potassium in liquid N-methylacetamide (NMA), a model of the main chain of amino acids, is equal to that of potassium in water (*Roux and Bernèche, 2002*). The solvation free energy of $K^+$ in liquid amide is not experimentally known, but by comparison with data available for similar organic solvents one would suggest a value of about −2 kcal/mol in reference to solvation in water (*Marcus et al., 1988*). Given the uncertainty on this value and that it is arguably small, it was decided in the context of the free energy calculations published in *Bernèche and Roux (2001)* to set the $K^+$ solvation free energy in liquid amide and water equal. For consistency with this previous work, we maintained the same parameters for the current study. Periodic boundary conditions were applied and long-range electrostatic interactions were calculated using the Particle Mesh Ewald algorithm (*Essmann et al., 1995*). The molecular systems were equilibrated for about 400 ps with decreasing harmonic restraints applied to the protein atoms, the ions and the water molecules localized in the P-loop and the filter. All trajectories were generated with a timestep of 2 fs at constant normal pressure (1 Atm) controlled by an extended Lagrangian algorithm (*Feller et al., 1995*) and constant temperature (323.25 K) using a Nose-Hoover thermostat (*Evans and Holian, 1985*). In the simulations of the open E71A constructs, the conformation of the intracellular gate was maintained by a harmonic root-mean-square-deviation (RMSD) restraints (force constant of 10 kcal/mol•Å with an offset of 0.1 Å) applied to residues 22 to 38 and 100 to 124. Trajectories of 20 ns were produced for each system (some in two replicas), and were used for the distance analyses (*Figure 2*). For the simulation-illustrating ion permeation (*Figure 5—figure supplement 3*), the E71A/L40A construct of the partially activated channel (pdb entry 3F7V) was used with an increase of the ion concentration to 800 mM KCl and the application of a transmembrane voltage of 400 mV across the simulation box along the Z-axis.

## Modeling the open activated selectivity filter

The computational method used to prepare the open state channel is designed to mimic ion-mediated recovery from the inactivated selectivity filter state, as shown in *Ostmeyer et al. (2013)*. Based on the 3F7V and 3F5W open inactivated structures, the collapsed selectivity filter was remodeled in the putative conducting conformation through equilibration runs of 400 ps in which water molecules were removed from the P-loop, and $K^+$ ions were maintained in state S1-S3-Cav or S0-S2-S4 using harmonic restraints of 10 kcal/mol•Å. To accelerate the transition from the collapsed to the conductive conformation, the NBFIX correction to the ion-carbonyl interactions was not applied for this first phase of the equilibration, as in *Ostmeyer et al. (2013)*. The systems were afterward further equilibrated by running simulations using the NBFIX correction to the ion-carbonyl interactions and without any restraint applied to the ions. The molecular systems obtained after 2 ns of simulations were used as initial coordinates for the PMF calculations.

## Potential of mean force calculations

The 2D potential of mean force (PMF) calculations were performed using the self-learning adaptive umbrella sampling method (*Wojtas-Niziurski et al., 2013*). Starting from a single initial configuration, this self-learning approach automatically constructs simulation windows following the valleys of lower free energy. PMF calculations were initiated from the occupancy states S1-S3-Cav and S0-S2-S4 as specified in the text. The upper free energy limit for the creation of new windows was set to 12 kcal/mol. The reaction coordinates were defined as the distance along the pore axis (which is aligned with the Z axis) between an ion, or the center-of-mass of two ions, and the center of mass of the selectivity filter backbone (residues 75 to 79). Independent simulations of 600 ps were performed every 0.5 Å along these reaction coordinates using a biasing harmonic potential with a force constant of 20 kcal/mol•Å$^2$ and were unbiased using the weighted histogram analysis method (WHAM) (*Kumar et al., 1992*) with a grid spacing of 0.1 Å in both dimensions. The first 100 ps of

sampling is considered as equilibration and is not included in the final PMF calculation. For each PMF, the reference free energy value is set to zero at the free energy minimum. The statistical error on the PMF calculations is calculated by subdividing the sampling in slices of 100 ps (*Figure 5—figure supplement 4*). In a subsequent step, a free energy off-set is attributed to each interval PMF according to a least-square fit to the final PMF considering only the grid points for which the free energy is <6 kcal/mol. Using the PMFs of the last five intervals as an ensemble, the standard deviation is calculated at every grid points (*Figure 1—figure supplement 4* and *Figure 5—figure supplement 5*).

The PMFs presented in *Figure 5* were obtained from the combined sampling of two independent automated umbrella-sampling simulations initiated in different ion occupancy states, respectively S1-S3-Cav and S0-S2-S4. The approach is similar to that routinely applied for combining forward- and backward-free energy perturbation simulations using the Bennett Acceptance Ratio (BAR) (*Lu et al., 2003*).

## Force field and ion permeation

At odds with the results presented here, PMF calculations presented in *Bernèche and Roux (2001)* described diffusion limited ion permeation in the KcsA selectivity filter despite a closed intracellular gate. These free energy simulations were performed based on the 3.2 Å structure of the closed KcsA channel (pdb entry 1BL8) (*Doyle et al., 1998*) using CHARMM22, an earlier generation of the CHARMM force field (*MacKerell et al., 1998*). Since then, correction terms were added to the $\Phi$, $\Psi$ backbone potential energy function in the CHARMM force field to better reproduce crystallographic data and quantum mechanic calculations (*Mackerell et al., 2004*). As documented by MacKerell and co-workers, these corrections, which are now included in the CHARMM36 force field (*Best et al., 2012*), led to a decrease of the backbone RMS fluctuations across all tested proteins (*Mackerell et al., 2004*). Thus, the CHARMM22 force field was biased toward higher backbone fluctuations, which favored ion permeation in simulations of the closed KcsA channel as reflected by the PMF calculations found in *Bernèche and Roux (2001)*. The CHARMM36 force field, on the other hand, results in more limited fluctuations of the KcsA selectivity filter that are not sufficient to sustain ion permeation in the closed channel. Repeating the PMF calculation on the 1BL8 structure using the latest force field yielded essentially the same results as those obtained with the 1K4C structure, and thus the differences between those structures are not critical (data not shown). The fluctuations required for ion permeation are recovered when the intracellular gate of the channel opens. The CHARMM22 force field allowed for a description of the ion permeation mechanism in the selectivity filter of K$^+$ channels. The CHARMM36 force field allows us to go further and explain how ion permeation is regulated.

Our findings differ from those of *Köpfer et al. (2014)* who proposed a hard knock-on mechanism in which the binding of 4 ions to the selectivity filter is required for permeation to take place in the open KcsA channel, versus 3 ions intercalated with water molecules in our simulations. Although Köpfer and colleagues also have used the CHARMM36 force field, they have not, unlike us, applied a NBFIX correction to the interaction between K$^+$ and backbone carbonyl oxygen atoms. This lead to a stronger K$^+$-carbonyl interaction in their CHARMM36 simulations compared to ours, with a difference in K$^+$ solvation-free energy in liquid amide of about 8 kcal/mol. For ion permeation to take place, the ion binding affinity to the selectivity filter needs to be compensated by the ion-ion electrostatic repulsion. Thus, in the systems of Koepfer et al. (2014)*Köpfer et al., 2014*, the intrinsically higher ion-binding affinity requires a higher number of bound ions to promote ion permeation, impeding at the same time the entrance of water molecules into the selectivity filter.

## Protein purification and reconstitution

The L40A mutation was made in a construct of the non-inactivating (*Cordero-Morales et al., 2006*) KcsA-E71A pQE60 by QuikChange mutagenesis (Agilent Technologies, Santa Clara, CA) and verified with sequencing. KcsA protein was expressed and purified as described previously (*Heginbotham et al., 1999*; *Thompson et al., 2008*; *Posson et al., 2013a*). Briefly, KcsA protein expression was induced in BL21 (DE3, Invitrogen, Carlsbad, CA) *E. coli* cells at 37°C in Luria-Bertani media with 500 µM IPTG. Cells were sonicated (Thermo Fisher Scientific, Waltham, MA) in a suspension (100 mM KCl, 50 mM Tris, pH 7.5) and KcsA was extracted with 25 mM n-decyl maltoside (DM,

Anatrace, Maumee, OH) and then purified from the clarified, soluble fraction in 100 mM KCl, 20 mM Tris, and 5 mM DM, pH 7.5 buffer using a $Ni^{2+}$-affinity column (Novagen, Merck, Germany) and gel filtration (Superdex 200, GE Healthcare, Chicago, IL).

KcsA was reconstituted into liposomes using protein to lipid ratios of 0.1–2.5 µg protein per mg of lipid (3:1 POPE: POPG, Avanti Polar Lipids, Alabaster, AL). Detergent was removed from protein-lipid mixtures using a gel filtration column (G50 fine, GE Healthcare) in buffer (400 mM KCl, 5 mM NMG, 20 mM Tris, pH 7.5). Liposome aliquots were flash-frozen in liquid nitrogen and stored at −80°C.

## Single-channel recording and analysis

KcsA L40A/E71A channels were recorded and analyzed as described previously (*Thompson et al., 2008*; *Posson et al., 2013a*). Briefly, channel liposomes were fused to horizontal planar lipid bilayers (3:1 POPE: POPG in decane) and single channel currents were obtained using 70 mM KCl, 30 mM KOH, 10 mM MOPS, 10 mM succinate, and 10 mM Tris, pH adjusted using HCl to pH 7 for the *cis* (extracellular side) and pH 4–6 for the *trans* (cytoplasmic side). Single channel recordings of the KcsA L40A/E71A mutant established channel activity (conductance and open probability) similar to the previously studied E71A control channel at pH 4. Rare channels where this was not the case were excluded from the analysis. Subsequent perfusion of the cytoplasmic side to higher pH values was used to determine the pH-dependent gating.

Bilayers were voltage-clamped using an Axopatch 200B amplifier (Molecular Devices, Sunnyvale, CA) and current recordings were filtered at 2 kHz using a 4-pole Bessel filter, digitized at 25 kHz using a Digidata 1440A (Molecular Devices), and recorded in Clampex 10. Single channel open probabilities ($P_o$) were calculated using Clampfit 10 (Molecular Devices) and Po vs. pH data were fit in Origin (OriginLab, Northampton, MA) with the Hill equation (*Equation 1*):

$$P_o = \frac{P_o^{\max}}{1 + (\frac{EC_{50}}{[H^+]})^{n_H}} = \frac{P_o^{\max}}{1 + (10^{(pH-pH_{1/2})})^{n_H}} \tag{1}$$

$P_o^{max}$ is the maximal open probability, $EC_{50}$ is the proton concentration of half-activation (also represented by $pH_{1/2}$), $[H^+]$ is the proton concentration (also represented by $pH$), and $n_H$ is the Hill coefficient.

The $P_o$ vs. $pH$ data were also fit to a two proton-binding sites MWC model (*Equation 2*) (*Posson et al., 2013a*):

$$P_o = \frac{L_o \left(1 + \frac{[H^+]}{K_{a1}^{open}}\right)^4 \left(1 + \frac{[H^+]}{K_{a2}^{open}}\right)^4}{L_o \left(1 + \frac{[H^+]}{K_{a1}^{open}}\right)^4 \left(1 + \frac{[H^+]}{K_{a2}^{open}}\right)^4 + \left(1 + \frac{[H^+]}{K_{a1}^{closed}}\right)^4 \left(1 + \frac{[H^+]}{K_{a2}^{closed}}\right)^4}$$

$$= \frac{L_o (1 + 10^{(pK_{a1}^{open}-pH)})^4 (1 + 10^{(pK_{a2}^{open}-pH)})^4}{L_o (1 + 10^{(pK_{a1}^{open}-pH)})^4 (1 + 10^{(pK_{a2}^{open}-pH)})^4 + (1 + 10^{(pK_{a1}^{closed}-pH)})^4 (1 + 10^{(pK_{a2}^{closed}-pH)})^4} \tag{2}$$

$L_o$ is the intrinsic gating equilibrium between open and closed in the absence of protons, $[H^+]$ is the proton concentration (also represented by $pH$), $K_a^{open}$ and $K_a^{closed}$ are the acid dissociation constants for a proton sensor in the open and closed states, respectively (also represented by $pK_a^{open}$ and $pK_a^{closed}$). *Equation 2* contains two proton sensors that are labeled with subscripts 1 and 2. For the analysis of L40A/E71A, independent adjustment of $L_o$ or pH-sensor $pK_a$ values were not sufficient to adequately describe the data. Since the mutant dose response was very similar to that observed in the H25R mutant (*Posson et al., 2013a*) (*Figure 4*), we assumed that L40A altered both Lo and $pK_a^{closed}$. Using these two free parameters, the best fit resulted in an increased $L_o$ value and a loss of the H25 pH-sensor because the $pK_a^{closed}$ value climbed to a constrained maximum value equal to $pK_a^{open}$. Subsequently constraining $pK_a^{closed}$ resulted in a fit for $L_o$, the single free parameter. Model parameters for L40A and relevant pH-sensor mutants (*Posson et al., 2013a*) are summarized in *Table 1*.

## Acknowledgements

FTH and SB are grateful to Wojciech Kopec and Bert L de Groot for constructive exchanges on ion parametrization and its impact on ion permeation. SB acknowledges support by the Swiss Foundation for Excellence and Talent in Biomedical Research, the Swiss National Science Foundation (SNF Professorship No PP00P3_139205) and the FP 7 European Union Human Brain Project (grant No 604102). CMN acknowledges support from NIH grant R01GM088352 and Irma T Hirschl trust. Calculations were performed at sciCORE (http://scicore.unibas.ch/) scientific computing core facility at University of Basel and at the Swiss National Supercomputing Centre (CSCS) under project ID s421.

## Additional information

### Funding

| Funder | Grant reference number | Author |
|---|---|---|
| Swiss National Science Foundation | SNF Professorship No. PP00P3_139205 | Simon Bernèche |
| FP7 European Union | Human Brain Project No. 604102 | Simon Bernèche |
| NIH Office of the Director | R01GM088352 | Crina M Nimigean |
| Swiss Foundation for Excellence and Talent in Biomedical Research | | Simon Bernèche |
| Irma T Hirschl Trust | | Crina M Nimigean |

The funders had no role in study design, data collection and interpretation, or the decision to submit the work for publication.

### Author contributions

Florian T Heer, Software, Formal analysis, Validation, Investigation, Visualization, Methodology, Writing—original draft; David J Posson, Formal analysis, Validation, Investigation, Methodology, Writing—original draft, Writing—review and editing; Wojciech Wojtas-Niziurski, Software, Formal analysis, Validation, Investigation, Visualization, Methodology; Crina M Nimigean, Conceptualization, Resources, Formal analysis, Supervision, Funding acquisition, Validation, Investigation, Methodology, Writing—original draft, Project administration, Writing—review and editing; Simon Bernèche, Conceptualization, Resources, Formal analysis, Supervision, Funding acquisition, Investigation, Visualization, Methodology, Writing—original draft, Project administration, Writing—review and editing

### Author ORCIDs

David J Posson http://orcid.org/0000-0002-6491-8238
Crina M Nimigean http://orcid.org/0000-0002-6254-4447
Simon Bernèche https://orcid.org/0000-0002-6274-4094

### Decision letter and Author response

Decision letter https://doi.org/10.7554/eLife.25844.020
Author response https://doi.org/10.7554/eLife.25844.021

## Additional files

### Supplementary files
• Transparent reporting form
DOI: https://doi.org/10.7554/eLife.25844.019

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
