## [Decision Letter]

Thank you for submitting your article "Mechanism of Activation at the Selectivity Filter of the KcsA K^+^ Channel" for consideration by *eLife*. Your article has been favorably evaluated by Richard Aldrich (Senior Editor) and three reviewers, one of whom, Kenton Swartz, is a member of our Board of Reviewing Editors. The reviewers have opted to remain anonymous.

The reviewers have discussed the reviews with one another and the Reviewing Editor has drafted this decision to help you prepare a revised submission.

Summary:

This is an interesting manuscript advancing a provocative new idea about the gating mechanism of tetrameric cation channels. Decades of work on various proteins in this family demonstrate that for many members, such as the Shaker Kv channel, the internal pore functions as a gate to regulate ion permeation. In these instances, the selectivity filter in the external pore determines the species of ions that can permeate and functions as an inactivation gate to limit ion permeation as the channel inactivates in the continual presence of an activating stimuli. For some members, such as BK and CNG channels, the internal pore undergoes a conformational change in response to the activating stimuli but seems to not function as a barrier to ion permeation; in these instances, the selectivity filter has been proposed to function as the primary gate. Multiple structures have been solved of the KcsA potassium channel in closed and open states, showing that the internal pore indeed opens in response to protonation and that the selectivity enters what appears to be an inactivated conformation. Previous simulations from this group and others have suggested that the selectivity filter is capable of supporting ion permeation in channels with a closed internal pore. Based on computation investigations of the KcsA potassium channel using improved force fields, the authors now propose that the structure of the selectivity filter is *not* capable of supporting ion permeation in channels with a closed internal pore, and that opening of the internal pore articulates a subtle structural change in the filter to a conducting state. The idea they advance is that the filter fluctuates between non-conducting closed to conducting open and then to non-conducting inactivated states, and that outward movement of the TM1 helix articulates the closed to open structural change in the filter. They further propose that L40 in TM1 directly interacts with S69 and V70 at the bottom of the pore helix to couple movements of the internal pore with the closed-open change in the filter. When viewed with a perspective of everything that is known about gating of ion access to the pores of different channels in the family of tetrameric cation channels, they advance the idea that it is the selectivity filter that functions as a universal gate in all channels, and although movement of the internal pore is required to open the filter, it need not actually close to prevent ion permeation. The ideas advanced in this manuscript are fascinating and if true would represent a major advance in our understanding of gating in the large and diverse family of tetrameric cation channels. The following are issues that the authors need to address in revision.

Essential revisions:

1) The functional characterization of the L40A mutant needs to be enhanced and more rigorously presented. The authors fit the Po-pH relationship with a specific model wherein Lo increases, from which they conclude that opening is favored. This is not very convincing when one sees kinetics like those shown at pH 5.5 where both open and closed times appear to decrease. The authors need to show more data, including traces and both open and closed time distributions at different pH values, and do a better job of articulating what exactly can be learned from their analysis. Can the results only be interpreted as an increase in Lo? The authors should also mention the caveat that which gate(s) is(are) opening and closing during the single channel recordings is unclear.

2) The authors should do a better job of describing their manipulations to the filter to make it conductive. The brief description in the second paragraph of the subsection “Relaxation of the selectivity filter is required for ion permeation”, says little and one must go to the methods for even a conceptual picture of what was done. How do we know the result is not some artificially widened or broken filter? We are told the filter is occupying space left open by the outward movement of L40 by TM1, but is there evidence for a stable open filter state, such as through time-dependent structural analysis?

3) It is essential that the authors present simulations that explicitly demonstrate permeation events can occur at a reasonable rate through the water-mediated knock-on mechanism depicted in Figure 1 – when the channel is in the activated state hypothesized here (i.e. both at the intracellular gate and the filter), and when a reasonable membrane voltage is applied across the membrane. We appreciated that the focus of this study is not the mechanism of K^+^ permeation per se; however, the fact is that authors consider the feasibility of permeation as the reporter of the activation state, logically. E.g. the filter is said to be inactive in the closed state of KcsA (Figure 1) because permeation appears unfeasible – for the specific mechanism considered in the PMF calculations. It is therefore key to show this is the relevant mechanism. We also appreciate that the PMF surface in Figure 1 is consistent with possibility of fast diffusion (in contrast to that in Figure 1), but this calculation does not explicitly demonstrate that permeation events will be observed, or that they will proceed primarily though the mechanism that is evaluated in the PMF calculations. Our request stems from the apparent discrepancy between the results presented here and those previously reported by Koepfer et al. in Science 2014, 346 p352 – who do focus on the mechanism of K^+^ permeation. In that study, the authors consider a hypothetically conducting state of KcsA that is based on an open structure of the channel (PDB 3f5w), but with a selectivity filter re-modeled as in the closed state (1k4c). Our understanding is that here, Heer et al. consider this same hypothetical state, although we imagine the structures are not identical to begin with. Koepfer et al. do not compute PMF surfaces, and instead design their simulations to evaluate the rate of ion permeation across the channel under an applied membrane voltage – without imposing a specific mechanism, aside from the starting condition. The results that are relevant here are those obtained with the CHARMM36 forcefield – which is also used here. At ~300 mV Koepfer et al. observe no permeation events through a water-mediated knock-on mechanism akin to that studied in Figure 1, where 2 ions occupy the filter (S1-S4 sites) at a time. Instead, Koepfer et al. report that permeation events occur when 4 ions occupy the filter, under larger voltages (~600 mV). The data and conclusions of this study implicit contradict the results of Koepfer et al., but the authors do not explicitly address this discrepancy – which, in our view, not only undermines the impact of this study, but will also add further confusion to the field. We therefore believe that it is essential that additional simulations be provided to more directly demonstrate that a water-mediated knock-on mechanism is indeed the primary permeation mechanism.

4) What is striking in this manuscript is the abandonment of the past Nature paper by Berneche and Roux 2001, with recomputed calculations yielding very different answers. The low (1-2 kcal/mol) barriers for ion movement in 2001 and high (7 kcal/mol) barriers here in 2017 are explained loosely in terms of improved force fields, with reference to supplementary materials (being only general) glossing over a critical issue. If changes in protein force field in CHARMM are responsible for a slight change in positional distribution sufficient to eliminate conduction, then it is vital that the accuracy of those new terms for the particular selectivity filter secondary structure be explained. The authors could also compare distributions (like Figure 1 and Figure 2) between existing old and new force field simulations to show the difference.

5) The identification of L40 in the communication of gate-filter change is interesting, but discussion of the subsequent simulations of this mutant focuses only on a partially activated state. The lacking effect on the closed state is dismissed. Surely, if L40 is important, the presumed non-conducting filter would become more flexible around S4 and reduce the barrier toward 2 kcal/mol. i.e. Why does Leu-Ala mutation not free up space behind the pore helix comparable to slight TM1 movement? Is it a matter of not sampling the change in short 20ns (or 600ns umbrella sampling) simulations?

6) The authors set K^+^ water and liquid amide solvation free energies equal. KCl partitioning involves a 3.5 kcal/mol shift (Yu et al. 2010. JACS. 132:10847), and we might expect some difference for K^+^? DE Shaw (Jensen et al. 2013. J. Gen. Physiol. 141:619), and more recently de Groot et al., have explored the roles of K-carbonyl parameters, perhaps warranting more explanation for the choice, which could determine the barriers that distinguish conduction states. If you change the K^+^ – C=O interaction enough, you will change the mechanism altogether. But even small changes can alter the maps presented. How do we know what is right?

[Editors' note: further revisions were requested prior to acceptance, as described below.]

Thank you for resubmitting your work entitled "Mechanism of Activation at the Selectivity Filter of the KcsA K^+^ Channel" for further consideration at *eLife*. Your revised article has been favorably evaluated by Richard Aldrich (Senior Editor) and three reviewers, one of whom, Kenton Swartz, is a member of our Board of Reviewing Editors.

The manuscript has been improved but there are some remaining issues that need to be addressed before acceptance, as outlined below:

The authors have done a good job revising the manuscript to address the reviewers' comments. As a result the manuscript has been greatly improved, and it is now considerably more accessible to the general reader. However there remain two concerns that must be addressed before the manuscript can be accepted for publication.

The first concern pertains to the length of the unbiased simulation (Figure 5—figure supplement 3) provided in support of the water-mediated knock-on mechanism analyzed in the free-energy landscapes. Given that the applied potential is fairly large (400 mV) it is a concern that this simulation might simply reflect a perturbation of the KWKWK configuration assumed initially. This concern would be addressed if the trajectory was extended to reveal the permeation of K^+^ ions not initially bound to the selectivity filter.

The second issue pertains to the free-energy landscapes shown in Figure 5 – which derive from the combination of the 'forward' and 'backward' calculations described in Figure 5—figure supplement 1. The extent to which these 'forward' and 'backward' free-energy surfaces differ from each other is problematic. This difference is arguably a good indicator of the statistical error of the calculations, and is typically much greater than the error inferred from the block analysis in Figure 5—figure supplement 5. It can be therefore questioned whether the simulation data supports the conclusion that the L40A mutation reduces the free-energy barriers between the two ion configurations considered. It would be important that the authors demonstrate that the magnitude of this change exceeds the magnitude of the error, either through additional analysis of the existing data or, if necessary, with additional simulations.

[Editors' note: further revisions were requested prior to acceptance, as described below.]

Thank you for resubmitting your work entitled "Mechanism of Activation at the Selectivity Filter of the KcsA K^+^ Channel" for further consideration at *eLife*. Your revised article has been favorably evaluated by Richard Aldrich (Senior Editor) and three reviewers, one of whom, Kenton Swartz, is a member of our Board of Reviewing Editors.

The manuscript has been improved but there are remaining issues that need to be addressed before acceptance, as outlined below:

The authors have made additions with extended unbiased simulations, and a response to the concern about hysteresis. We will not discuss further the limited unbiased simulations as we believe they demonstrate, in a limited way, that conduction might occur with water mediation. But concern remains about the calculations to observe the effect of L40A mutation, which should be addressed, or at least presented in a way that more transparently reflects the possible uncertainties in those calculations.

The data in Figure 5 attempted to show that L40A mutation influences PMF barriers, and is thus important to the story. The authors volunteered separate simulations with S0-S2-S4 and S1-S3-Cav starting points, yielding free energy maps that differ by up to 7 kcal/mol, with very different shapes. Yet the authors just combined these trajectories into a single WHAM solution to yield Figure 5. When questioned about this, the authors added unconvincing statements in the caption to suggest that this has nothing to do with error. They just cite a past study that shows combination of different endpoint free energies using Bennett Acceptance Ratio (BAR) can lead to improvement (and note that their theory WHAM and BAR become the same in the limit of infinitesimal bin size; though that is not the case here). While it might be credible, in general, that one can extract some meaningful data from such combination of disparate histograms, it is misleading to write in this fashion, as that study has not been demonstrated to have similar extents of endpoint distribution overlap that could allow judgment of error here. The statements come across as if it has been shown for this current problem that it would not lead to error, which is not the case. In fact there is *no* guarantee that one can combine any endpoint calculations and expect the combination to be meaningful. If the authors are to argue this line, then they should be obliged to carry out analysis to demonstrate the resultant errors are indeed small, as currently they remain undetermined.

We have limited faith in the quantitative nature of the combined results in Figure 5. Without demonstration of the magnitude of the errors in the combined WHAM calculations, we cannot put a measure of confidence in the results for the effects of mutation. This is not the main result of the paper, but it is one interesting component. We suggest that the misleading comments be removed and replaced with statements about likely uncertainties based on the supplementary figure results.

---

## [Author Response]

Essential revisions:1) The functional characterization of the L40A mutant needs to be enhanced and more rigorously presented. The authors fit the Po-pH relationship with a specific model wherein Lo increases, from which they conclude that opening is favored. This is not very convincing when one sees kinetics like those shown at pH 5.5 where both open and closed times appear to decrease. The authors need to show more data, including traces and both open and closed time distributions at different pH values, and do a better job of articulating what exactly can be learned from their analysis. Can the results only be interpreted as an increase in Lo? The authors should also mention the caveat that which gate(s) is(are) opening and closing during the single channel recordings is unclear.

We would like to thank the reviewers for this suggestion. We reworked Figure 4 to now include 3 sets of single-channel traces for comparison between the control channel, the L40A, and the H25R mutant channels. It is now easy to see that the control channel is closed at pHs ranging from 7 to 5.5 and opens very cooperatively with protons over ~0.5 pH units, where the activity goes from nearly 0 to 100%. Indeed, it is difficult to identify a recording where the control channel is captured at an intermediate pH where it displays an intermediate open probability. In contrast, L40A opens at lower proton concentrations than the control channel, and has much less cooperative opening, as evidenced by the gradually increasing activity displayed at intermediate pH conditions. This is manifested in a decreased Hill slope for L40A compared with the control, as shown in Figure 4.

Thus, proton binding is less coupled to L40A channel opening, to an extent mimicking that of a KcsA mutant with an extreme phenotype where one of two pH sensors was mutated, H25R, with single channel records shown side by side in Figure 4 and dose response in Figure 4. The simplest way to interpret the behavior of both mutants in the context of a model we previously proposed in Posson et al., 2014, is to increase L0, since the channels require fewer protons to open and are open at intermediate levels of pH. The results cannot only be interpreted as a decrease in L0 since the decrease in slope also requires less coupling between proton binding and channel opening, and was best (and most parsimoniously) fit with only one weak pH sensor, as opposed to the two (one strong, one weak) required by the control channel. It is of course possible to fit L40A dose-response with more complex models, for instance involving two weak pH sensors instead of one. However, since we do not have any information about the degree of involvement of the individual sensors in the gating of this mutant and we are not drawing any conclusions based on the number of pH sensors remaining, we did not elaborate on these alternative models and stuck with the most parsimonious model required to fit the data.

We believe that kinetic analysis of KcsA gating is not necessary for an appreciation of the impact the L40A mutation has on channel function, as described above. Since we did not perform any kinetic analysis of either the control channel or L40A, we have removed all reference to gating kinetics from the paper. The significance that position L40 has on KcsA gating can now be easily understood using the proton dose-responses and the sets of single-channel traces of the control and mutant channels, L40A and H25R (Figure 4). All these properties have been described with an improved text in the third paragraph of the subsection “The outer helix is involved in activation gating”.

We added a sentence in the figure legend of Figure 4 where we state that we don’t know the identity of the gates that open and close during the single-channel recording traces shown.

2) The authors should do a better job of describing their manipulations to the filter to make it conductive. The brief description in the second paragraph of the subsection “Relaxation of the selectivity filter is required for ion permeation”, says little and one must go to the methods for even a conceptual picture of what was done. How do we know the result is not some artificially widened or broken filter? We are told the filter is occupying space left open by the outward movement of L40 by TM1, but is there evidence for a stable open filter state, such as through time-dependent structural analysis?

The computational method used to prepare the open state model is designed to mimic ion-mediated recovery from the collapsed or inactivated selectivity filter state, as shown in Ostmeyer et al. 2013 (1). We believe that this approach, which does not involve direct restraints on the selectivity filter itself, minimizes the possibility of introducing structural artifacts into the channel.

We have improved the description of the open state model preparation as follows: “Two KcsA structures with an open bundle-crossing and collapsed selectivity filter (pdb entries 3F7V and 3F5W, with intracellular entryway diameters of 23 Å and 32 Å, respectively) were chosen for the preparation of simulations of an open KcsA channel. […] This procedure led to a selectivity filter nearly identical in structure with that of the closed state KcsA (Figure 1—figure supplement 2).” Further details are provided in a new subsection “Modeling the open activated selectivity filter” in Materials and Methods.

The molecular representations in Figure 1 and Figure 1—figure supplement 2 show that the selectivity filter can locally enlarge to accommodate ions and water molecules in intermediate states, but overall maintains a canonical conformation. A time-series presented in Figure 5—figure supplement 3 shows that the channel is stable on the time scale of many tens of nanoseconds.

1) Ostmeyer J., Chakrapani S., Pan A.C., Perozo E., and Roux B. Recovery from slow inactivation in K^+^ channels is controlled by water molecules.” Nature 501:121–124, 2013.

3) It is essential that the authors present simulations that explicitly demonstrate permeation events can occur at a reasonable rate through the water-mediated knock-on mechanism depicted in Figure 1 – when the channel is in the activated state hypothesized here (i.e. both at the intracellular gate and the filter), and when a reasonable membrane voltage is applied across the membrane. We appreciated that the focus of this study is not the mechanism of K^+^ permeation per se; however, the fact is that authors consider the feasibility of permeation as the reporter of the activation state, logically. E.g. the filter is said to be inactive in the closed state of KcsA (Figure 1) because permeation appears unfeasible – for the specific mechanism considered in the PMF calculations. It is therefore key to show this is the relevant mechanism. We also appreciate that the PMF surface in Figure 1 is consistent with possibility of fast diffusion (in contrast to that in Figure 1), but this calculation does not explicitly demonstrate that permeation events will be observed, or that they will proceed primarily though the mechanism that is evaluated in the PMF calculations. Our request stems from the apparent discrepancy between the results presented here and those previously reported by Koepfer et al. in Science 2014, 346 p352 – who do focus on the mechanism of K^+^ permeation. In that study, the authors consider a hypothetically conducting state of KcsA that is based on an open structure of the channel (PDB 3f5w), but with a selectivity filter re-modeled as in the closed state (1k4c). Our understanding is that here, Heer et al. consider this same hypothetical state, although we imagine the structures are not identical to begin with. Koepfer et al. do not compute PMF surfaces, and instead design their simulations to evaluate the rate of ion permeation across the channel under an applied membrane voltage – without imposing a specific mechanism, aside from the starting condition. The results that are relevant here are those obtained with the CHARMM36 forcefield – which is also used here. At ~300 mV Koepfer et al. observe no permeation events through a water-mediated knock-on mechanism akin to that studied in Figure 1, where 2 ions occupy the filter (S1-S4 sites) at a time. Instead, Koepfer et al. report that permeation events occur when 4 ions occupy the filter, under larger voltages (~600 mV). The data and conclusions of this study implicit contradict the results of Koepfer et al., but the authors do not explicitly address this discrepancy – which, in our view, not only undermines the impact of this study, but will also add further confusion to the field. We therefore believe that it is essential that additional simulations be provided to more directly demonstrate that a water-mediated knock-on mechanism is indeed the primary permeation mechanism.

As requested, we have performed simulations describing K^+^ permeation through the open KcsA channel with an applied transmembrane voltage of 400 mV and a K^+^ concentration of 800mM. Because it is central to our study, the E71A/L40A construct was used for these simulations. A time-series analysis is presented in Figure 5—figure supplement 3. The trajectory shows that water molecules are intercalated between ions in all permeation events, in agreement with the PMF calculations presented in the paper. We added text describing these results in the last paragraph of the subsection “The outer helix is involved in activation gating”.

We appreciate the reviewers’ concerns about the great interest and possible confusion resulting from the differences in permeation mechanism seen in our simulations and those presented in Koepfer et al., Science 2014. While it is beyond the scope of the present work to examine the root cause of these differences (this is indeed an ongoing effort across several groups), we have added the following additional text: “Departing from the permeation mechanism described here which involves the binding of three ions to the selectivity filter, Koepfer et al. (2014) suggested a hard knock-on mechanism in which ion permeation arise from the entrance of a 4th ion, excluding water molecules from the selectivity filter. This discrepancy between the two proposed permeation mechanisms is in part due to difference in the K^+^ ion force field parameters, which results in much stronger ion binding affinity to the selectivity filter in the study by Koepfer et al. (see subsection “Force field and ion permeation” in Materials and methods).” This paragraph is followed by a discussion related to point 6 below.

We also added a more technical discussion in the subsection “Force field and ion permeation” in Materials and methods: “Our findings differ from those of Koepfer et al. (2014) who proposed a hard knock-on mechanism in which the binding of 4 ions to the selectivity filter is required for permeation to take place in the open KcsA channel, versus 3 ions intercalated with water molecules in our simulations. […] Thus, in the systems of Koepfer et al., the intrinsically higher ion binding affinity requires a higher number of bound ions to show ion permeation, impeding at the same time the entrance of water molecules into the selectivity filter.”

While Koepfer et al. have used several different force fields we have limited our discussion to their simulations based on the CHARMM force field, which are representative of their general conclusions. The apparent discrepancy between different simulation studies and the divergent permeation mechanisms should be addressed more exhaustively in future studies.

4) What is striking in this manuscript is the abandonment of the past Nature paper by Berneche and Roux 2001, with recomputed calculations yielding very different answers. The low (1-2 kcal/mol) barriers for ion movement in 2001 and high (7 kcal/mol) barriers here in 2017 are explained loosely in terms of improved force fields, with reference to supplementary materials (being only general) glossing over a critical issue. If changes in protein force field in CHARMM are responsible for a slight change in positional distribution sufficient to eliminate conduction, then it is vital that the accuracy of those new terms for the particular selectivity filter secondary structure be explained. The authors could also compare distributions (like Figure 1 and Figure 2) between existing old and new force field simulations to show the difference.

To improve our explanation about this important issue, we have updated the text in the fifth paragraph of the Results subsection “Relaxation of the selectivity filter is required for ion permeation”, as well as in the subsection “Force fields and ion permeation” in Materials and methods. This essentially is as follows.

The simulations published in Bernèche and Roux, 2001 were performed with the CHARMM22 force field. Since then, correction terms were added to the 𝜙, 𝜓 backbone potential energy function in the CHARMM force field to better reproduce crystallographic data and quantum mechanic calculations (2). As documented by MacKerell and co-workers, these corrections, which are now included in the CHARMM36 force field (3), led to a decrease of the backbone RMS fluctuations across all tested proteins (2). Thus, the CHARMM22 force field was biased toward higher backbone fluctuations, which was favoring ion permeation in K channels as reflected by the PMF calculations found in Bernèche and Roux, 2001. The CHARMM36 force field describes better the backbone fluctuations and shows that in the closed state of the KcsA channel, the fluctuations of the selectivity filter are not sufficient to sustain ion permeation. The fluctuations required for permeation are recovered when the intracellular gate of the channel opens up. The CHARMM22 force field allowed us to describe the ion permeation mechanism in the selectivity filter of K channels. The CHARMM36 force field allows us to go further and explain how ion permeation is regulated.

Following the reviewers’ suggestion to provide a comparison of the selectivity filter fluctuations between the older and newer versions of the force field, we added to Figure 2 data extracted from simulations published in Bernèche and Roux (2001), which were based on the CHARMM22 force field. The plot shows that, using the CHARMM22 force field, the fluctuations of the selectivity filter in the closed channel are slightly larger than the fluctuations observed in the open channel with CHARMM36.

2) MacKerell Jr. A.D., Feig M., and Brooks III C.L. "Extending the treatment of backbone energetics in protein force fields: limitations of gas-phase quantum mechanics in reproducing protein conformational distributions in molecular dynamics simulations." Journal of Computational Chemistry, 25: 1400-1415, 2004.

3) Best R.B., Zhu X., Shim J., Lopes P.E.M., Mittal J., Feig M., and MacKerell Jr. A.D. "Optimization of the additive CHARMM all-atom protein force field targeting improved sampling of the backbone phi, psi and side-chain chi1 and chi2 dihedral angles." Journal of Chemical Theory and Computation, 8: 3257-3273, 2012.

5) The identification of L40 in the communication of gate-filter change is interesting, but discussion of the subsequent simulations of this mutant focuses only on a partially activated state. The lacking effect on the closed state is dismissed. Surely, if L40 is important, the presumed non-conducting filter would become more flexible around S4 and reduce the barrier toward 2 kcal/mol. i.e. Why does Leu-Ala mutation not free up space behind the pore helix comparable to slight TM1 movement? Is it a matter of not sampling the change in short 20ns (or 600ns umbrella sampling) simulations?

We agree with the reviewers that, according to our hypothesis, the L40A mutation would be expected to produce a visible effect on the energetics of ion permeation in the closed channel. After analysis of the free energy data, we considered the possibility of a hysteresis, i.e. a dependency on the starting ion occupancy state of the PMF calculations, which could obfuscate the mechanism. We thus ran new PMF calculations starting from the S0-S2-S4 occupancy state for both the control channel and the L40A mutant in the closed (PDB ID 1K4C) and the partially activated (PDB ID 3F7V) conformations. The PMF calculations shown in the original manuscript were initiated in occupancy state S1-S3-Cav. The forward and backward umbrella sampling data were combined and yielded new PMFs, which we present in Figure 5. The effect of the L40A mutant is now clearly visible in the closed channel, as the free energy barriers between stable occupancy states are lower, and new stable states are made accessible in comparison to the control channel. In the partially activated conformation, the effect is less strong but it is still visible. This is expected since the opening of the channel reduces the steric interactions between residue L40 and the pore helix, and thus the impact of the L40A mutation is reduced. The discussion on page 9 was modified accordingly. The individual forward and backward PMFs are presented in Figure 5—figure supplement 1, and show that the L40A mutation facilitates ion permeation in all cases, except one that is non-conclusive. We believe these results are more comprehensive than those presented in the original version of the manuscript and we thank the reviewers for requesting that we address this issue.

6) The authors set K^+^ water and liquid amide solvation free energies equal. KCl partitioning involves a 3.5 kcal/mol shift (Yu et al. 2010. JACS. 132:10847), and we might expect some difference for K^+^? DE Shaw (Jensen et al. 2013. J. Gen. Physiol. 141:619), and more recently de Groot et al., have explored the roles of K-carbonyl parameters, perhaps warranting more explanation for the choice, which could determine the barriers that distinguish conduction states. If you change the K^+^ – C=O interaction enough, you will change the mechanism altogether. But even small changes can alter the maps presented. How do we know what is right?

The solvation free energy of K^+^ in liquid amide has not been experimentally determined. By comparison with data available for K^+^ in other similar organic solvents (4), the solvation free energy of K^+^ in liquid amide was estimated to be -2 kcal/mol different from solvation of K^+^ in water. Given that the value is uncertain and arguably small, it was decided in the context of the free energy calculations published in 2001 to set the K^+^ solvation free energies in liquid amide and water equal (Bernèche and Roux, Nature 2001). For consistency with this previous work, we maintained the same parameters for the current study. A mention to this effect was added to the Materials and methods subsection “Molecular simulation systems” (last paragraph).

Regarding the concern reviewers have toward the effects of changing the K^+^-carbonyl interaction parameters, previous simulations performed by Jensen et al. (2013) based on the open Kv1.2/2.1 channel show that the permeation mechanism is robust and is not dictated by the ion force field. They reported that the outcome of changing the parameters defining the interaction between K^+^ ions and carbonyl groups (i.e. by either incorporating the NBFIX correction mentioned above or leaving it out) is a change in the relative probabilities of observing the two permeation pathways described in Bernèche and Roux (2001) and in the current manuscript, i.e. the knock-on and vacancy diffusion pathways. These pathways alternate between 2-ion and 3-ion occupancy states, with intercalated water molecules. Thus, the activated state of the channel, which has the lowest ion binding affinity, is not susceptible to the effect of the parameters defining the K^+^-carbonyl interactions.

Similarly, as we demonstrated in this manuscript using the weakest K^+^-carbonyl interaction (i.e. using the NBFIX correction), the closed state of the KcsA selectivity filter presents the highest ion binding affinity, and further strengthening of the K^+^-carbonyl interactions by removing the NBFIX correction would simply increase further the ion binding affinity and prevent permeation. Thus, we are confident that the choice of K^+^-carbonyl parameters, within the range of values previously used, does not affect the interpretation of the mechanism we proposed here, for either the open or closed states of K^+^ channels.

It remains unclear why, in the study of de Groot and colleagues, permeation in the open KcsA channel is susceptible to the ion force field (see point 3 above), while, in the study of Jensen et al., permeation in the Kv1.2/2.1 chimera is not. One possibility is that the truncated structure of the open KcsA is not as stable as the Kv1.2/2.1 structure and that small, uncontrolled reorientation of the TM helices in KcsA could partially close the selectivity filter and increase its ion binding affinity. This increase of ion binding affinity combined with an ion force field that defines stronger K^+^-carbonyl interactions could result in a system in which the selectivity filter can bind four K ions. In order to prevent such uncontrolled conformational changes of the open intracellular gate in KcsA, we applied a rmsd restraint to residues on the intracellular side of TM1 and TM2 (see Materials and methods).

This discussion is summarized as follows at the end of the paragraph related to point 3 above: “[…] it is important to consider that, as shown here, the ion binding affinity is essentially determined by the functional state of the selectivity filter. […] It remains unclear why permeation in the open KcsA channel is susceptible to the ion force field while permeation in the Kv1.2/2.1 chimera is not (Jensen 2013, Koepfer 2014).”

4) Marcus Y., Kamlet M.J., and Taft R.W. “Linear Solvation Energy Relationships. Standard Molar Gibbs Free Energies and Enthalpies of Transfer of Ions from Water into Nonaqueous Solvents.” Journal of Physical Chemistry,92: 3613-3622, 1988.

[Editors' note: further revisions were requested prior to acceptance, as described below.]

The manuscript has been improved but there are some remaining issues that need to be addressed before acceptance, as outlined below:The authors have done a good job revising the manuscript to address the reviewers' comments. As a result the manuscript has been greatly improved, and it is now considerably more accessible to the general reader. However there remain two concerns that must be addressed before the manuscript can be accepted for publication.The first concern pertains to the length of the unbiased simulation (Figure 5—figure supplement 3) provided in support of the water-mediated knock-on mechanism analyzed in the free-energy landscapes. Given that the applied potential is fairly large (400 mV) it is a concern that this simulation might simply reflect a perturbation of the KWKWK configuration assumed initially. This concern would be addressed if the trajectory was extended to reveal the permeation of K^+^ ions not initially bound to the selectivity filter.

We have extended the simulation to 200 ns. The trajectory shows that an incoming ion at t= 2 ns leaves the pore around t= 120 ns. In the interval t= 40 ns to 200 ns, the selectivity filter undergoes two full cycles starting with ions in S2-S4, followed by incoming ions around t= 45 ns and 105 ns, the release of ions at t= 45 ns and 120 ns, and the return to state S2-S4 around t= 50 ns and 190 ns. Note that the S2-S4 state was not imposed and emerges from the simulation, first at t= 40 ns.

The second issue pertains to the free-energy landscapes shown in Figure 5 – which derive from the combination of the 'forward' and 'backward' calculations described in Figure 5—figure supplement 1. The extent to which these 'forward' and 'backward' free-energy surfaces differ from each other is problematic. This difference is arguably a good indicator of the statistical error of the calculations, and is typically much greater than the error inferred from the block analysis in Figure 5—figure supplement 5. It can be therefore questioned whether the simulation data supports the conclusion that the L40A mutation reduces the free-energy barriers between the two ion configurations considered. It would be important that the authors demonstrate that the magnitude of this change exceeds the magnitude of the error, either through additional analysis of the existing data or, if necessary, with additional simulations.

We disagree with the view that the difference between the forward and backward free energy surfaces is a good indication of the statistical error. It was shown for free energy perturbation calculations that combining forward and backward sampling is improving the accuracy of the calculation and that the statistical error is much less than that inferred from the difference between the forward and backward simulations (Lu et al., 2003). Combining the forward and backward simulations is done using algorithms such as the Multistate Bennett Acceptance Ratio (mBAR) or the Weighted Histogram Analysis Method (WHAM) – mBAR corresponds to WHAM with a bin-width approaching zero. The same approach is here applied to potential of mean simulations.

We have added the following to the caption of Figure 5—figure supplement 1:

“In presence of the L40A mutation, free energy barriers are reduced under all four conditions. In the forward calculation of the closed channel, the L40A mutant impacts mainly on the transition between the S1-S3 and S2-S4 states, while in all other conditions the mutation also reduces the free energy barrier for a third ion to bind or unbind (transition toward or from state S0-S2-S4). […] The statistical error on the PMFs of Figure 5 was estimated using block averaging (see Figure 5—figure supplement 4 and Figure 5—figure supplement 5).”

[Editors' note: further revisions were requested prior to acceptance, as described below.]

The manuscript has been improved but there are remaining issues that need to be addressed before acceptance, as outlined below:The authors have made additions with extended unbiased simulations, and a response to the concern about hysteresis. We will not discuss further the limited unbiased simulations as we believe they demonstrate, in a limited way, that conduction might occur with water mediation. But concern remains about the calculations to observe the effect of L40A mutation, which should be addressed, or at least presented in a way that more transparently reflects the possible uncertainties in those calculations.The data in Figure 5 attempted to show that L40A mutation influences PMF barriers, and is thus important to the story. The authors volunteered separate simulations with S0-S2-S4 and S1-S3-Cav starting points, yielding free energy maps that differ by up to 7 kcal/mol, with very different shapes. Yet the authors just combined these trajectories into a single WHAM solution to yield Figure 5. When questioned about this, the authors added unconvincing statements in the caption to suggest that this has nothing to do with error. They just cite a past study that shows combination of different endpoint free energies using Bennett Acceptance Ratio (BAR) can lead to improvement (and note that their theory WHAM and BAR become the same in the limit of infinitesimal bin size; though that is not the case here). While it might be credible, in general, that one can extract some meaningful data from such combination of disparate histograms, it is misleading to write in this fashion, as that study has not been demonstrated to have similar extents of endpoint distribution overlap that could allow judgment of error here. The statements come across as if it has been shown for this current problem that it would not lead to error, which is not the case. In fact there is no guarantee that one can combine any endpoint calculations and expect the combination to be meaningful. If the authors are to argue this line, then they should be obliged to carry out analysis to demonstrate the resultant errors are indeed small, as currently they remain undetermined.We have limited faith in the quantitative nature of the combined results in Figure 5. Without demonstration of the magnitude of the errors in the combined WHAM calculations, we cannot put a measure of confidence in the results for the effects of mutation. This is not the main result of the paper, but it is one interesting component. We suggest that the misleading comments be removed and replaced with statements about likely uncertainties based on the supplementary figure results.

Following the reviewers’ recommendations we have rearranged Figure 5 and its associated supplements. Figure 5—figure supplement 4 and 5 now present the statistical error of the individual forward and backward calculations instead of being based on the combined sampling, which was problematic.

We believe that the combined PMFs provides a valuable and convenient summary of the complex picture depicted by the individual forward and backward simulations shown in Figure 5—figure supplement 1, and thus we kept Figure 5 as it was. However, we changed Figure 5 caption to indicate how these PMFs were calculated and that the statistical error cannot be readily determined for the combined sampling.

The individual PMFs presented in Figure 5—figure supplement 1 show that, no matter what initial state is used, the mutation favors transitions between the 2-ion (S1-S3-Cav and S2-S4-Cav) and 3-ion (S0-S2-S4) states by either reducing the free energy difference (see closed channel) or the free energy barriers (see partially activated channel) between these states.

The combined PMFs of Figure 5 lead to the same conclusions. While the statistical error on these PMFs would remain to be clarified, the PMFs are provided as a summary of the plots in Figure 5—figure supplement 1, which *all* show reduced free energy barriers.

We wanted to keep the visual summary in Figure 5, so we explain how the PMFs were obtained to avoid any misunderstanding, and then send the reader to figure supplement 1 for more details and to figure supplement 4 and 5 for the statistical errors. In the end, both Figure 5 and Figure 5—figure supplement 1 lead us to the same conclusions.

We added the following to the main text:

“The free energy calculations presented in Figure 5 combine data from independent automated umbrella sampling simulations initiated in different ion occupancy states, as detailed in Figure 5—figure supplement 1. The PMFs of both Figure 5 and Figure 5—figure supplement 1 lead to the following observations. In the closed conformation (1K4C) of the control channel, the state with 3 ions bound to the filter (S0-S2-S4) is overly stabilized in comparison to states with only 2 ions bound to the filter (S1-S3-Cav and S2-S4-Cav), with a free energy difference of about 11 kcal/mol. Free energy barriers of 6 to 10 kcal/mol are observed between the 2-ion and 3-ion states. The E71A/L40A mutant brings about fluctuations that reduce the ion binding affinity and the relative stability of the 3-ion state. The 2-ion states are stabilized by about 6 kcal/mol, and are thus more accessible.[…].”

We also modified the captions of Figure 5 and Figure 5—figure supplement 1 as follows:

“Figure 5.[…] [W]ith a free energy difference between the 3-ion (S0-S2-S4) and the 2-ion (S1-S3-Cav and S2-S4-Cav) states of about 11 kcal/mol, and free energy barriers of up to 9 kcal/mol between these states. The L40A mutation stabilizes the 2-ion states by about 6 kcal/mol and reduces the free energy barriers to 4-5 kcal/mol. […] Each of the 4 PMFs shown here combines together data from two independent automated umbrella-sampling simulations initiated in states S1-S3-Cav and S0-S2-S4, respectively. Since the statistical error on the PMFs combined in this fashion might not be readily definable, we only present the statistical errors associated with the underlying individual PMFs (Figure 5—figure supplement 1 (individual PMFs), 4 (convergence) & 5 (statistical error)).[…] Despite this limitation, the combined PMFs shown here provide a convenient summary of the underlying calculations presented separately in Figure 5—figure supplement 1.”

“Figure 5—figure supplement 1.[…]In the case of the closed channel (A), the main impact of the L40A mutation is to make accessible the states in which only 2 ions are bound to the selectivity filter (states S1-S3-Cav and S2-S4-Cav). The closed structure has such a high ion binding affinity that it over-stabilizes the state with three ions bound to the filter (S0-S2-S4). The L40A mutation adds fluctuations that reduce the binding affinity and the relative stability of the 3-ion state. In the partially activated channel (B), the main impact of the mutation is to reduce the free energy barrier along the permeation pathway […]”.

Finally, the following was added to the Materials and methods, in the subsection “Potential of Mean Force Calculations”:

“The PMFs presented in Figure 5 are obtained from the combined sampling of two independent automated umbrella-sampling simulations initiated in different ion occupancy states, respectively S1-S3-Cav and S0-S2-S4. The approach is similar to that routinely applied for combining forward and backward free energy perturbation simulations using the Bennett Acceptance Ratio (BAR) (Lu et al., 2003).”